# How Do Augmentations with Label Smoothing Enhance Model Robustness?

## Abstract

Model robustness indicates a model's capability to generalize well on unforeseen distributional shifts, including data corruption, adversarial attacks, and domain shifts. One of the most prevalent and effective ways to enhance the robustness often involves data augmentations and label smoothing techniques. Despite the great success of the related approaches in diverse practices, a unified theoretical understanding of their efficacy in improving model robustness is lacking. We offer a theoretical framework to clarify how augmentations, label smoothing, or their combination enhance model robustness through the lens of loss surface flatness, generalization bound, and adversarial robustness. Specifically, we first formally bridge the diversified data distribution via augmentations to the flatter minima on the parameter space, which directly links to the improved generalization capability. Moreover, we further bridge augmentations with label smoothing, which softens the confidence of the target label, to the improved adversarial robustness. We broadly confirm our theories through extensive simulations on the existing common corruption and adversarial robustness benchmarks based on the CIFAR and *tiny*ImageNet datasets, as well as various domain generalization benchmarks.

## 1 Introduction

*Model robustness*, which is seen as a critical factor of deep models in applications requiring high reliability such as autonomous vehicles and medical diagnosis, entails maintaining performance despite data distribution shifts. In the past decade, *data augmentation* has been widely used as a popular and pragmatic technique to enhance the model performance, as well as the robustness against data corruption, adversarial attacks, or even domain shifts (DeVries & Taylor, 2017; Zhang et al., 2018; Hendrycks et al., 2021a; Cubuk et al., 2019; Xu et al., 2023). The intuition of its efficacy relies on the belief that augmentations enrich the training data distribution, which allows models to easily extrapolate to unseen data distributions, which is the so-called generalization capability.

On the other hand, *label smoothing*, which softens the confidence of target labels, has been studied for its robustness benefits, particularly against adversarial perturbations (Szegedy et al., 2016; Guo et al., 2017). When combining label smoothing with augmentations such as adversarial perturbations, also known as adversarial training, it has shown promise in strengthening model robustness against adversarial attacks (Shafahi et al., 2019; Ren et al., 2022).

Despite the strong utility of augmentations, label smoothing, or their combinations, there is a lack of unified theories that formally clarify how they generally enhance model robustness. The prior formal analyses are quite limited on handling a few types of augmentations and addressing adversarial robustness. As one of the existing analyses, adversarial perturbation with label smoothing is proved to improve the smoothness of loss values on the input space, especially along the direction of adversarial perturbation (Ren et al., 2022). However, the theory is still limited in the adversarial settings, failing to elaborate the settings of the general form of augmentations and the generalization capability against unforeseen distributional shifts of data samples. As another analysis for other types of augmentations, Mixup (Zhang et al., 2018) has been proved to enlarge the boundary thickness, which is the marginal space between two differently labeled samples that is believed to improve the adversarial robustness (Yang et al., 2020). Still, the analysis is restricted to mixup-based augmentations and lacks a rigorous link to the improved performance on adversarial samples.

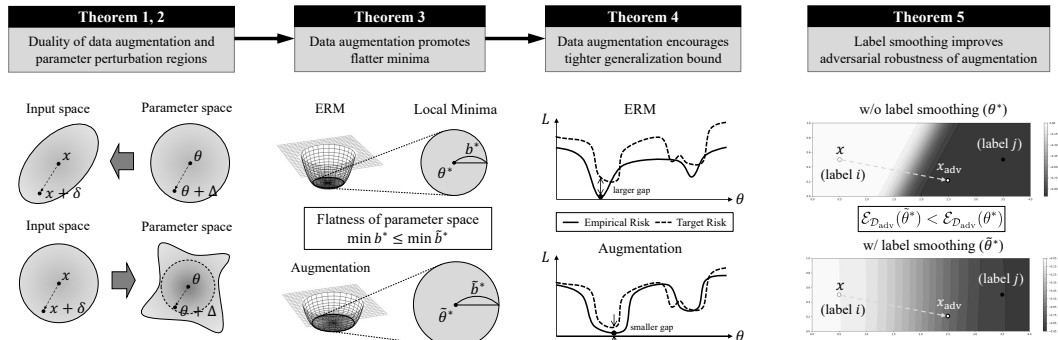

Figure 1: A sketch of the links between augmentations with label smoothing and model robustness.

In this paper, we first provide theories for clarifying how augmentations combined with label smoothing can bolster model robustness. Our analysis has two main logical sequences: **i)** First, we formally bridge the general form of augmentations to the improved generalization bound. To give a brief sketch of our thinking, we start to demonstrate the duality between the input space region covered by the augmented samples and the corresponding parameter space region with the same loss values (formalized by **Theorem 1** and **Theorem 2**). Based on the duality, we then claim that the augmentations flatten the loss surface on the parameter space (**Theorem 3**), finally reaching to the improved generalization bound against distribution shifts via leveraging the flatten loss surface (**Theorem 4**). *These findings still remain valid when combined with label smoothing.* **ii)** Second, we switch the focus to the effect of label smoothing on adversarial robustness. As a rigorous link to the model performance against adversarial settings, we theoretically show that label smoothing lowers the adversarial risk of the models trained with augmentations. Fig. 1 shows a full sketch.

To provide empirical evidence of the claims, we strictly demonstrate the flatness of loss surface and the enhanced generalization performance through augmentations[1] In addition, we further demonstrate the improved adversarial robustness through label smoothing[2].

## 2 PRELIMINARIES

We provide the basic notations for further formulations and the background of model flatness.

### 2.1 BASIC NOTATIONS

Let us consider an input $x \in \mathbb{R}^n$ from input space $\mathcal{X}$, which is paired with a target label $y \in \mathbb{R}^c$ from label space $\mathcal{Y}$, where $n$ and $c$ are the dimensions of the input space and the label space. A model $f(\cdot; \theta) : \mathbb{R}^n \to \mathbb{R}^c$ parameterized by $\theta \in \mathbb{R}^m$ maps a given input to the estimated label, where $m$ is the dimension of the model parameter space. Based on the model, loss function $\mathcal{L}(\cdot, \cdot)$ is used for computing the loss value for a given data sample $(x, y)$ with model $f(\cdot; \theta)$ as follows: $\mathcal{L}(f(x; \theta), y)$. The true risk over the data distribution $\mathcal{D}$ can be formulated as follows:

$$\mathcal{E}_{\mathcal{D}}(\theta) := \mathbb{E}_{(x,y) \sim \mathcal{D}}\big[\mathcal{L}(f(x; \theta), y)\big]. \tag{1}$$

Similarly, empirical risk $\hat{\mathcal{E}}_{\mathcal{D}}(\theta)$ over $N$ samples from $\mathcal{D}$ is:

$$\hat{\mathcal{E}}_{\mathcal{D}}(\theta) := \frac{1}{N} \sum_{i=1}^{N} \mathcal{L}(f(x_i; \theta), y_i). \tag{2}$$

#### 2.1.1 DATA AUGMENTATION

Data augmentation $\mathcal{A}(\cdot) : \mathbb{R}^n \to \mathbb{R}^n$ augments a given input $x$ to augmented input $\tilde{x} := \mathcal{A}(x)$. Let us further represent it with the difference between the original input and the augmented one:

---

[1]Testing is done on domain generalization benchmarks (PACS, VLCS, OfficeHome) and common corruption benchmarks (CIFAR-10/100-C, *tiny*ImageNet-C), addressing domain shifts and data corruptions in images.

[2]Testing is done on adversarial benchmarks with the CIFAR-10/100, and *tiny*ImageNet datasets.

$\tilde{x} := \mathcal{A}(x) = x + \delta$, where $\delta \in \mathbb{R}^n$.[3] Also, $\mathcal{P}_{\mathcal{A}}(\tilde{x}|x)$ is the probability density function of augmented samples $\tilde{x}$, given $x$.

### 2.1.2 Label smoothing

Label smoothing $\mathcal{S}(\cdot) : \mathbb{R}^c \to \mathbb{R}^c$ is a process of softening hard label $y$ to obtain smoothed label $\tilde{y} := \mathcal{S}(y)$ as follows: $\tilde{y} := (1 - \epsilon) \cdot y + \epsilon/c \cdot \mathbf{1}$, where $0 < \epsilon < 1$ and $\mathbf{1}$ is a vector with ones at all elements, whose length is $c$. Label smoothing assigns the target label a probability of $1 - \epsilon$ and evenly distributes the remaining probability among the other classes.

### 2.2 Data augmentation with label smoothing

For a data sample $(x, y)$ from the original dataset $\mathcal{D}$, let us define $\tilde{\mathcal{A}}(x, y) := (\mathcal{A}(x), \mathcal{S}(y)) = (\tilde{x}, \tilde{y})$ as the augmented sample with label smoothing. Also, let us define $\tilde{\mathcal{D}} := \{(\tilde{x}, \tilde{y}) \mid \tilde{x} = \mathcal{A}(x), \tilde{y} = \mathcal{S}(y), (x, y) \sim \mathcal{D}\}$ as the augmented dataset with label smoothing.

### 2.3 Model Flatness

#### 2.3.1 Definition

*Model flatness* characterizes the extent of change in the model's loss values across proximate points in the parameter space. When the loss rapidly changes around the found minima, it indicates that the model is located at *sharp minima*. Otherwise, it denotes *flat minima* when the loss varies smoothly. The change of losses around the model parameters can be formalized as follows:

$$\max_{\|\Delta\| \leq \gamma} \mathbb{E}_{(x,y) \sim \mathcal{D}} \big[ \mathcal{L}(f(x; \theta + \Delta), y) - \mathcal{L}(f(x; \theta), y) \big], \tag{3}$$

where $\Delta \in \mathbb{R}^m$ is the perturbation around model parameter $\theta$ that maximally increases loss within a radius $\gamma > 0$.

#### 2.3.2 Sharpness-aware minimization

The most popular principal way for finding flatter minima is Sharpness-Aware Minimization (SAM) (Foret et al., 2021), which formally transforms loss minimization into a min-max optimization:

$$\min_{\theta} \max_{\|\Delta\| \leq \gamma} \mathbb{E}_{(x,y) \sim \mathcal{D}}[\mathcal{L}(f(x; \theta + \Delta), y)], \tag{4}$$

As formulated in the work of (Foret et al., 2021), when adding the loss value at $\theta$, only the first term of equation 3 remains, yielding the objective function in equation 4. Through the minimization of the maximization term, SAM aims to find the minima $\theta$ with flatter loss surfaces around $\gamma$ radius.

## 3 How Do Augmentations Enhance Generalization?

In this section, we provide a rigorous link from the data augmentations to the improved generalization capability. Our main claims are twofold: **i)** Translation between the manifolds on input and parameter spaces, **ii)** Association of flatter loss in the input space via augmentations with flatness in the parameter space, leading to a reduced generalization gap.

### 3.1 Translation Between Input and Parameter Spaces

Let us imagine a perturbation $\delta \in \mathbb{R}^n$ around the given input $x \in \mathbb{R}^n$. Our basic intuition is then that there exists a corresponding perturbation $\Delta \in \mathbb{R}^m$ around the parameter $\theta$, where the loss remains consistent:

$$f(x + \delta; \theta) = f(x; \theta + \Delta). \tag{5}$$

Beyond that, for all perturbation within a closed sphere $\|\delta\| \leq \gamma$ on input space, our intuition is that there exists the corresponding manifold of perturbations $\mathcal{R}_{\Theta}^{\gamma}$ around the parameter $\theta$, where the loss remains the same.

---

[3]By formulating augmentation as an additive perturbation, we transform the perturbation on the input space to the one on the parameter space. Details are in the following section.

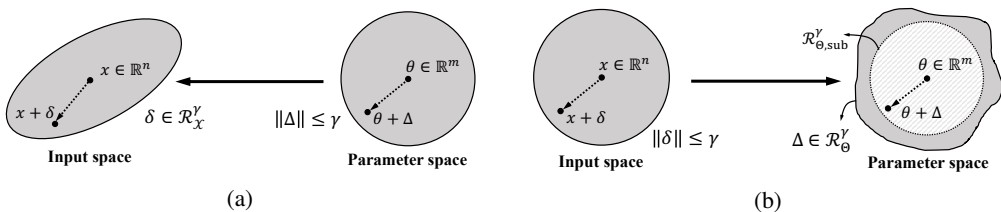

Figure 2: A graphical illustration of Theorem 1 (a) and 2 (b).

Reversely, for all perturbation within a closed sphere $\|\Delta\| \leq \gamma$ on parameter space, our idea is that there exists the corresponding manifold of perturbations $\mathcal{R}_\mathcal{X}^\gamma$ around the input $x$, where the loss remains the same.

Let us then formally define the manifolds $\mathcal{R}_\Theta^\gamma$ and $\mathcal{R}_\mathcal{X}^\gamma$.

**Definition 1.** *(For $\mathcal{R}_\mathcal{X}^\gamma$) Given $x \in \mathbb{R}^n$ and $\theta \in \mathbb{R}^m$, for perturbation bounded with $L_2$-norm of $\gamma$ on parameter space, i.e., $\{\Delta \in \mathbb{R}^m \mid \|\Delta\| \leq \gamma\}$, $\mathcal{R}_\mathcal{X}^\gamma \subset \mathbb{R}^n$ is the input space region satisfying the following two constraints:*

$$\text{For all } \|\Delta\| \leq \gamma, \text{ there exists } \delta \in \mathcal{R}_\mathcal{X}^\gamma \text{ s.t. equation 5 holds.}$$
$$\text{For all } \delta \in \mathcal{R}_\mathcal{X}^\gamma, \text{ there exists } \|\Delta\| \leq \gamma \text{ s.t. equation 5 holds.}$$

**Definition 2.** *(For $\mathcal{R}_\Theta^\gamma$) Given $x \in \mathbb{R}^n$ and $\theta \in \mathbb{R}^m$, for perturbation bounded with $L_2$-norm of $\gamma$ on input space, i.e., $\{\delta \in \mathbb{R}^n \mid \|\delta\| \leq \gamma\}$, $\mathcal{R}_\Theta^\gamma \subset \mathbb{R}^m$ is the parameter space manifold satisfying the following two constraints:*

$$\text{For all } \|\delta\| \leq \gamma, \text{ there exists } \Delta \in \mathcal{R}_\Theta^\gamma \text{ s.t. equation 5 holds.}$$
$$\text{For all } \Delta \in \mathcal{R}_\Theta^\gamma, \text{ there exists } \|\delta\| \leq \gamma \text{ s.t. equation 5 holds.}$$

For an arbitrary deep architectures of $f(\cdot; \theta)$, formalizing the existence of the regions $\mathcal{R}_\mathcal{X}^\gamma$ and $\mathcal{R}_\Theta^\gamma$ is intractable. We here narrow down our focus on a linear model to explicitly formalize the existence of the regions. For this case, the number of parameters, which is $m$, equals to $c \times n$.

Briefly, the region on the input space, i.e., $\mathcal{R}_\mathcal{X}^\gamma$, can be found as an exact solution (referred to **Theorem 1**). Reversely, the region on parameter space, i.e., $\mathcal{R}_\Theta^\gamma$, can be bounded with the subset as follows: $\mathcal{R}_{\Theta,\text{sub}}^\gamma \subseteq \mathcal{R}_\Theta^\gamma$, which proves the existence of the region (referred to **Theorem 2**).

**Theorem 1.** *(A closed-form of $\mathcal{R}_\mathcal{X}^\gamma$) Given $x \in \mathbb{R}^n$ and $\theta \in \mathbb{R}^{c \times n}$, for perturbation bounded with $L_2$-norm of $\gamma$ on parameter space, i.e., $\{\Delta \in \mathbb{R}^{c \times n} \mid \|\Delta\| \leq \gamma\}$, $\mathcal{R}_\mathcal{X}^\gamma$ is a $c$-dimensional rotated ellipsoid centered at $x$:*

$$\mathcal{R}_\mathcal{X}^\gamma = \{\delta \in \mathbb{R}^n \mid \delta^\top U^\top D U \delta \leq 1\}, \tag{6}$$

*where $U$ is from Singular Value Decomposition the model parameter $\theta$ with a form of $(c \times n)$ matrix: $\theta = U\Sigma V^\top$. Also, $D$ is the $(c \times c)$ diagonal matrix whose $i$-th diagonal element is $\sigma_i \|x\| \gamma$, where $\sigma_i$ is the $i$-th singular value of $\theta$.*

**Remark 1.1.** *(A ball on parameter space is translated to a rotated ellipsoid on input space) The loss values inside a ball centered at the parameter $\theta$ directly correspond to the loss values of inputs inside the rotated ellipsoid $\mathcal{R}_\mathcal{X}^\gamma$ around $x$. When the model shows flatter loss surface around $\theta$, it indicates that the input region around $x$ with a shape of ellipsoid shows the flatter loss behavior.*

Fig. 2a illustrates the translation based on Theorem 1. The proof is provided in Appendix B.1.

**Theorem 2.** *(A subset of $\mathcal{R}_\Theta^\gamma$, i.e., $\mathcal{R}_{\Theta,\text{sub}}^\gamma$) Given dataset $\mathcal{D} = \{x_i\}_{i=1}^N$ and $\theta \in \mathbb{R}^{c \times n}$, for perturbation bounded with $L_2$-norm of $\gamma$ on input space, i.e., $\{\delta \in \mathbb{R}^n \mid \|\delta\| \leq \gamma\}$, $\mathcal{R}_{\Theta,\text{sub}}^\gamma$ is the subset of $\mathcal{R}_\Theta^\gamma$, i.e., $\mathcal{R}_{\Theta,\text{sub}}^\gamma \subseteq \mathcal{R}_\Theta^\gamma$:*

$$\mathcal{R}_{\Theta,\text{sub}}^\gamma = \{\Delta \in \mathbb{R}^{c \times n} \mid \|\Delta\| \leq \gamma^2 \sigma_{\min}^2 / \|x_{\max}\|^2\}, \tag{7}$$

*where $\sigma_{\min}^2 := \min_i \sigma_i^2$ is the minimum singular value of $\theta$ and $x_{\max} := \max_{x \in \mathcal{D}} \|x\|$.*

**Remark 2.1.** *(A ball on input space is translated to the region containing a ball on parameter space) The loss values inside a ball centered at the input $x$ directly correspond to the loss values for model parameters inside the ball $\mathcal{R}_\Theta^\gamma$, which contains the subset of region $\mathcal{R}_{\Theta,\text{sub}}^\gamma$. The claim shows that the region $\mathcal{R}_\Theta^\gamma$ indeed exists. Moreover, when the model shows a flatter loss behavior around $x$, it indicates that the parameter ball around $\theta$ shows a flatter surface, i.e., a flat minimum.*

Fig. 2b illustrates the translation based on Theorem 2. The proof is provided in Appendix B.2. For the extension to the case of deep architectures, we provide further discussions in Appendix D.2.

## 3.2 Linking Augmentations to Model Flatness and Generalization

Built upon the theorems above, we now formalize how can augmentations lead to flatter loss landscape around the minima and the improved generalization capability. Before providing the details, let us rephrase the formal definition of flat minima, which is called $b$-flat minima (Shi et al., 2021):

**Definition 3.** *($b$-flat local minima) Given loss $\mathcal{L}(\cdot, \cdot)$ and dataset $\mathcal{D}$, a model parameter $\theta \in \mathbb{R}^m$ is $b$-flat minima if the followings hold for the perturbation on parameter $\Delta \in \mathbb{R}^m$:*

$$\text{For all } \|\Delta\| \leq b, \ \hat{\mathcal{E}}_\mathcal{D}(\theta + \Delta) = \hat{\mathcal{E}}_\mathcal{D}(\theta) \text{ and there exists } \|\Delta\| > b, \ \hat{\mathcal{E}}_\mathcal{D}(\theta + \Delta) > \hat{\mathcal{E}}_\mathcal{D}(\theta). \tag{8}$$

Before describing $b$-flat minima, let us assume a property of $\mathcal{P}_\mathcal{A}(\tilde{x}|x)$ for the augmentation.

**Assumption 1.** *Given $x \in \mathbb{R}^n$ and augmentation $\mathcal{A}(\cdot)$, the probability density function of $\mathcal{P}_\mathcal{A}(\tilde{x}|x)$ satisfies:*

$$\text{For all } \|\delta\| \leq \gamma, \ \mathcal{P}_\mathcal{A}(\tilde{x}|x) > 0, \tag{9}$$

*where $\delta = \tilde{x} - x$ and $\gamma$ is some positive real number.*

From the assumption, the distribution of the augmented samples covers the region around the given original input $x$. Let $\Theta^*$ and $\tilde{\Theta}^*$ be the sets of the optimal model parameters whose elements $\theta^* \in \Theta^*$ and $\tilde{\theta}^* \in \tilde{\Theta}^*$ satisfy the following equalities, $\mathcal{E}_\mathcal{D}(\theta^*) = 0$ and $\mathcal{E}_{\tilde{\mathcal{D}}}(\tilde{\theta}^*) = 0$, respectively. Then our claim is that the minimum $b$-flatness among the solution parameters in $\tilde{\Theta}^*$, shows large $b$ (flatter) than the minimum $b$-flatness among the solutions in $\Theta^*$, which is trained on $\mathcal{D}$:

**Theorem 3.** *(Flatness of $\tilde{\theta}^*$) Let $\theta^* \in \Theta^*$ and $\tilde{\theta}^* \in \tilde{\Theta}^*$ be $b^*$ and $\tilde{b}^*$-flat minima, respectively. The following inequality holds:*

$$\min_{\theta^* \in \Theta^*} b^* \leq \min_{\tilde{\theta}^* \in \tilde{\Theta}^*} \tilde{b}^*. \tag{10}$$

**Remark 3.1.** *(Dense augmentations around the original sample encourage flatter minima) The key understanding of the theorem above is that the augmentations densely covers the sphere-shaped region around the given original sample with radius $\gamma$, then the model $\tilde{\theta}^*$ suppresses the loss values of the region. Next, the flat-region on the input space is translated to the closed region on parameter space, i.e., $\mathcal{R}_\Theta^\gamma$, which at least contains $\mathcal{R}_{\Theta,\text{sub}}^\gamma$ (referred to **Theorem 2**).*

The proof is provided is in Appendix B.3.

The final linkage to the generalization capability is straightforward by relying on the prior theoretical results that bridge Robust Risk Minimization (RRM) and the generalization bound (Cha et al., 2021). RRM aims to find a flat minimum, where its surrounding region shows flat loss surface. Based on Theorem 3, augmentations make the optimal parameter locate at flatter surface and it implies the improved generalization bound (referred to **Theorem 4**).

**Theorem 4.** *(Generalization bound) Given $M$ covering sets $\{\Theta_k\}_{k=1}^M$ of parameter space $\Theta$ with $\Theta = \bigcup_{k=1}^M \Theta_k$ and $diam(\Theta) = \sup_{\theta, \theta' \in \Theta} ||\theta - \theta'||_2$, where $M = \left\lceil \frac{diam(\Theta)}{\gamma} \right\rceil^d$ (d is the dimension of $\Theta$ and $\gamma$ is the value satisfying the condition of Assumption 1), and VC dimension $v_k$ for each $\Theta_k$, the following inequality holds with probability at least $1 - \delta$:*

$$\mathcal{E}_\mathcal{T}(\theta) < \hat{\mathcal{E}}_{\tilde{\mathcal{D}}}(\theta) + \frac{1}{2}\boldsymbol{Div}(\mathcal{D}, \mathcal{T}) + \log c + \max_{k \in [1, M]} \left[ \sqrt{\frac{v_k \log(N/v_k)}{N} + \frac{\log(M/\delta)}{N}} \right],$$

*where $\boldsymbol{Div}(\mathcal{D}, \mathcal{T}) = 2 \sup_A |\mathbb{P}_\mathcal{D}(A) - \mathbb{P}_\mathcal{T}(A)|$ measures the maximal discrepancy between distributions $\mathcal{D}$ and $\mathcal{T}$, and $N$ is the number of samples drawn from $\tilde{\mathcal{D}}$.*

Table 1: Flatness metrics of ERM and different augmentations on CIFAR-100 benchmark. (↓: *The lower the better, i.e., indicating flatter minimum*)

| Metrics | ERM | AugMix | PixMix | RandAug | StyleAug | CutOut |
|---|---|---|---|---|---|---|
| $\lambda_{\max} \downarrow$ | 12.04 | 2.25 $(-9.79)$ | 4.36 $(-7.68)$ | 2.49 $(-9.55)$ | 10.89 $(-1.15)$ | 1.94 $(-10.10)$ |
| Trace(H) $\downarrow$ | 540.89 | 110.42 $(-430.47)$ | 241.29 $(-299.60)$ | 131.65 $(-409.24)$ | 397.15 $(-143.74)$ | 100.59 $(-440.30)$ |
| $\mu_{\text{PAC-Bayes}} \downarrow$ | 164.50 | 98.22 $(-66.28)$ | 106.94 $(-57.56)$ | 101.63 $(-62.87)$ | 100.15 $(-64.35)$ | 95.47 $(-69.03)$ |
| LPF $\downarrow$ | 1.85 | 0.16 $(-1.69)$ | 0.26 $(-1.59)$ | 0.18 $(-1.67)$ | 0.20 $(-1.65)$ | 0.15 $(-1.70)$ |
| $\epsilon_{\text{sharp}} \downarrow$ | 17.25 | 9.71 $(-7.54)$ | 11.96 $(-5.29)$ | 10.54 $(-6.71)$ | 15.53 $(-1.72)$ | 8.53 $(-8.72)$ |

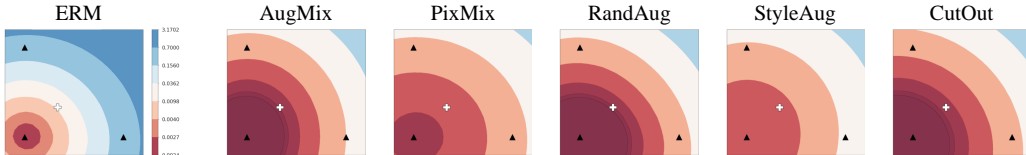

Figure 3: Loss surface visualization of ERM and augmentations on the CIFAR-100 benchmark. *(The wider red-colored region, i.e., flatter minima, the better.)*

**Remark 4.1.** *(**Augmentations improve generalization against data distribution shifts**) The theorem implies that the minimization of empirical loss for the augmented dataset, i.e., $\hat{\mathcal{E}}_{\tilde{\mathcal{D}}}(\theta)$, directly aims the tighter generalization bound on target distribution $\mathcal{T}$. Also, the term $M$ is related to $\gamma$, which measures how augmented data distribution covers wider range (referred to **Assumption 1**). When augmentation covers wider range around the original sample, i.e., a larger $\gamma$, it suppresses $M$, leading to the smaller last term of generalization bound. These interpretation elucidates how augmentations enhance the generalization capability against unseen data distribution shifts.*

The proof is provided is in Appendix B.4.

We want to emphasize that Theorem 1 to 4 are still remain valid with application of label smoothing. The reason is that label smoothing simply softens the hard label into a soft label, which does not involve the input and parameter translation via Theorem 1 and 2, and does not hurt the flatness and generalization bound via Theorem 3 and 4.

### 3.3 EMPIRICAL EXAMINATION ON MODEL FLATNESS AND GENERALIZATION

We evaluate how augmentations effect model flatness and generalization capability.

#### 3.3.1 AUGMENTATION METHODS TO BE CONSIDERED

There exists a massive number of augmentation methods in the literature. To evaluate the practical impact of well-known augmentation methods, we run experiments with a wide range of augmentation types and focus on advanced methods rather than a baseline. For the *model-free augmentation* type, we pick **CutOut** (DeVries & Taylor (2017)), which tunes a single image by cutting out a partial region of the image, and we also select **AugMix** (Hendrycks et al. (2021b)) and **PixMix** (Hendrycks et al. (2022)), which are advanced forms of trying to mix multiple clean images. For the *model-based augmentation* type, we pick **StyleAug** (Jackson et al. (2019)), which utilizes a generative model-based approach to diversify the style of clean images. Finally, among the *policy-based augmentation* type, we consider **RandAugment** (Cubuk et al. (2020)), which learns the policy for augmenting images.

#### 3.3.2 TESTS ON MODEL FLATNESS

**Flatness metrics:** To this end, we focus on measuring various quantitative flatness metrics, including the maximum eigenvalue of the Hessian matrix (i.e., $\lambda_{\max}$), the trace of the Hessian (i.e., Trace($\mathbf{H}$)), the PAC-Bayesian measure (i.e., $\mu_{\text{PAC-Bayes}}$) (Jiang et al., 2020), and the sharpness of the local minimum (i.e., $\epsilon_{\text{sharp}}$) (Keskar et al., 2017). Finally, we test Low-Pass Filter (LPF), which is recently suggested to show the robust correlation to generalization (Keskar et al., 2017). As a quali-

Table 2: Mean corruption error (mCE) of ERM and different augmentations on common corruption benchmarks, including CIFAR-10/100-C and tinyImageNet-C. (↓: *The lower the better*)

| Benchmarks | ERM | AugMix | PixMix | RandAug | StyleAug | CutOut |
|---|---|---|---|---|---|---|
| CIFAR-10-C ↓ | 25.57 | 12.47 ($-13.10$) | 8.70 ($-16.04$) | 17.83 ($-7.74$) | 18.91 ($-6.66$) | 24.25 ($-1.32$) |
| CIFAR-100-C ↓ | 52.21 | 38.71 ($-13.53$) | 33.08 ($-18.45$) | 44.68 ($-7.53$) | 49.01 ($-3.20$) | 51.84 ($-0.37$) |
| *tiny*ImageNet-C ↓ | 74.58 | 63.47 ($-11.11$) | 61.51 ($-13.57$) | 66.95 ($-7.63$) | 79.31 ($+4.73$) | 69.51 ($-5.07$) |
| **Average** ↓ | **50.78** | **38.22** ($-$**12.56**) | **34.43** ($-$**16.35**) | **43.15** ($-$**7.63**) | **48.52** ($-$**2.26**) | **48.53** ($-$**2.25**) |

Table 3: Domain generalization accuracies of ERM and different augmentations for domain generalization benchmarks, including PACS, VLCS, and OfficeHome. *(↑: The higher the better)*

| Benchmarks | ERM | AugMix | PixMix | RandAug | StyleAug | CutOut |
|---|---|---|---|---|---|---|
| PACS ↑ | 81.5 | 83.2 ($+1.7$) | 83.0 ($+1.5$) | 83.3 ($+1.8$) | 85.2 ($+3.7$) | 83.3 ($+1.8$) |
| VLCS ↑ | 77.0 | 78.0 ($+1.0$) | 77.8 ($+0.8$) | 78.6 ($+1.6$) | 77.6 ($+0.6$) | 78.6 ($+1.6$) |
| OfficeHome ↑ | 66.4 | 67.3 ($+0.9$) | 70.2 ($+3.8$) | 67.3 ($+0.9$) | 67.3 ($+0.9$) | 66.9 ($+0.5$) |
| **Average** ↑ | **75.0** | **76.2** ($+$**1.2**) | **77.0** ($+$**2.0**) | **76.4** ($+$**1.4**) | **76.7** ($+$**1.7**) | **76.3** ($+$**1.3**) |

tative result, we visualize the loss surface on the parameter space to illustrate how the learned models show a flatter loss landscape. We add the detailed descriptions of each metric in Appendix C.1.1.

**Evaluation results:** We trained WideResNet-40-2 architecture on CIFAR-100 datasets by adopting the five aforementioned augmentations. With the models, we present the statistics of the five selected flatness metrics in Table 1. As a baseline, we trained a model without augmentations, referred to as Empirical Risk Minimization (ERM). For all the augmentations, we observe the clear and consistently flatter minima of the models by augmentation methods over ERM. The empirical evidence directly coincides with Theorem 3, which proves the flatter surface of the models trained with augmentations. In Fig. 3, we visualize the loss landscape around the minima of models with augmentations. When compared to the loss surface of ERM, which shows sharper behavior, the models with augmentations exhibit a flatter loss landscape around the minima.

### 3.3.3 TESTS ON GENERALIZATION CAPABILITY

**Benchmarks to be considered:** Generalization capability implies a model ability to outstretch its performance to unseen data distributions. The meaning of 'unseen' can be widely varying from a naive train-test splitting, corruptions on clean samples, to some drastic shifts, including domain shifts and adversarial attacks. For various augmentation, we believe that a naive testing on novel samples without any meaningful data distribution shifts is already confirmed by prior works. We thus focus on rather severe data shifts, such as **common corruption**, including the CIFAR-10-C, the CIFAR-100-C, and the *tiny*ImageNet-C benchmarks (Hendrycks & Dietterich, 2019), and **domain generalization**, including PACS (Asadi et al., 2019), VLCS (Albuquerque et al., 2021), and OfficeHome (Zhou et al., 2020) benchmarks.

**Evaluation results:** As done in the flatness tests, we trained the same WRN-40-2 architecture on CIFAR-10-C and CIFAR-100-C by following Hendrycks et al. (2021b; 2022). For *tiny*ImageNet-C, we used ResNet18 by following Wang et al. (2021). As shown in Table 2, augmentations lead to the improved robustness to common corruptions with the consistent gains over ERM. It is empirical evidence of Theorem 4, which argues the tigher generalization bound via augmentations. For the case of *tiny*ImageNet-C with StyleAug, we conjecture that StyleAug adopts aggressive perturbations via jumping to other domains; it is not well-tailored to the additive corruptions of the benchmarks.

For the domain generalization testing, we follow the benchmarks proposed by DomainBed (Gulrajani & Lopez-Paz, 2021), utilizing the ResNet50 architecture as implemented in the framework. In Table 3, we empirically confirm that the augmentations show meaningful gains over ERM even in the domain shifts. It is noteworthy that StyleAug yields a much larger gain in the PACS case. We believe that the diversified 'styles' via StyleAug directly affects to the image-style-based domains of the PACS dataset, i.e., 'Photo', 'Cartoon', 'Art Painting', and 'Sketch'. Also, we emphasize that even though the augmentations are not tailored to the particular domain shifts, they leads to the clear

gains of generalization capability. It is a clear evidence of our main claim, i.e., Theorem 4. The details of the datasets and benchmarks are in Appendix C.1.2.

In Appendix C.2.2, we further discuss how augmented samples that are well-distributed around the original and satisfying Assumption 1 can relate to performance gains. Briefly, StyleAug and CutOut, which are slightly inferior to others, only weakly adhere to the assumption.

## 4 How Do Augmentations with Label Smoothing Enhance the Adversarial Robustness?

In this section, we explore how combining data augmentations with label smoothing can enhance a model's robustness against adversarial attacks.

### 4.1 Linking Augmentations with Label Smoothing to Adversarial Robustness

To understand this enhancement, we first define the adversarial dataset $\mathcal{D}_{\text{adv}}$ and its corresponding risk. Given a sampled dataset $\mathcal{D} = \{(x_i, y_i)\}_{i=1}^N$ and a classifier $f(\cdot; \theta)$, we follow the description in Carlini et al. (2019) to define the adversarial dataset as:

$$\mathcal{D}_{\text{adv}} := \{(x_{\text{adv}}, y) \mid x_{\text{adv}} = x + \delta_{\text{adv}}, (x, y) \in \mathcal{D}, f(x_{\text{adv}}; \theta) \neq y\}. \tag{11}$$

, where $\delta_{\text{adv}}$ is an adversarial perturbation making the samples cross the decision boundary with some constraints (e.g. $\|\delta_{\text{adv}}\| \leq h$ for some positive number $h$.) The adversarial empirical risk is formulated as $\hat{\mathcal{E}}_{\text{adv}}(\theta) := \frac{1}{N} \sum_{i=1}^N \left[ \mathcal{L}(f(x_i; \theta), y_i) \right]$, where $(x_i, y_i) \in \mathcal{D}_{\text{adv}}$. For the sake of simplicity, we regard $\mathcal{L}$ as a cross-entropy loss in this section.

Let $\tilde{\theta}_{LS}^*$ and $\tilde{\theta}^*$ be the optimal parameters of classifier $f$ that achieve zero loss of augmented data samples, *with* label smoothing and *without* label smoothing, respectively. Our claim shows the superiority of the model parameterized by $\tilde{\theta}_{LS}^*$ over $\tilde{\theta}^*$ in terms of a lower expected loss on $\mathcal{D}_{\text{adv}}$.

**Theorem 5.** *(Adversarial robustness) For the dataset of adversarial samples $\mathcal{D}_{\text{adv}}$, the models parameterized by $\tilde{\theta}_{LS}^*$ and $\theta^*$ satisfy the following inequality:*

$$\hat{\mathcal{E}}_{\mathcal{D}_{\text{adv}}}(\tilde{\theta}_{LS}^*) < \hat{\mathcal{E}}_{\mathcal{D}_{\text{adv}}}(\tilde{\theta}^*). \tag{12}$$

**Remark 5.1.** *(**Augmentations with the label smoothing enhance adversarial robustness**) We emphasize that applying label smoothing to augmentations enhances the model robustness against adversarial samples. To the best of our knowledge, the claim first proves the empirical gains observed by prior methods of augmentations with label smoothing in adversarial robustness.*

The proof for Theorem 5 is provided in Appendix B.5.

### 4.2 Empirical Examination on Adversarial Robustness

In this section, we demonstrate how label smoothing enhances adversarial robustness when applied to various augmentations. Using the same five target augmentations and model architecture from Section 3, we test model robustness against untargeted $L_\infty$ Projected Gradient Descent (PGD) attacks (Madry et al., 2018) on the CIFAR-10, CIFAR-100, and *TinyImageNet* datasets. As shown in Table 4, all augmentations, when combined with label smoothing, exhibit meaningful robustness gains over their original versions without label smoothing, clearly indicating improved resistance to adversarial attacks. This provides empirical support for Theorem 5, which asserts the benefit of label smoothing in enhancing adversarial robustness.

Further results are included in Appendix C.2.1, including adversarial error (complementing Table 4), model performance on PGD $L_2$ attacks, performance on clean datasets, and adversarial training (AT). All these results exhibit the same tendency.

## 5 Related Works

We here briefly categorize the related works into two parts: **i)** Existing data augmentations and label smoothing strategies, **ii)** Prior investigations of the impact of augmentations and label smoothing on model robustness, particularly from theoretical and empirical perspectives.

Table 4: Cross-entropy losses of different augmentations and their combination with label smoothing against adversarial perturbations from PGD $L_\infty$ untargeted attacks. 'Original' means the loss with original augmentation without label smoothing; '+ Label Smoothing (LS)' indicates the loss with label smoothing applied ($\downarrow$: *The lower the better*).

| Aug. | CIFAR-10 w/ $L_\infty \downarrow$ | | CIFAR-100 w/ $L_\infty \downarrow$ | | *tiny*ImageNet w/ $L_\infty \downarrow$ | | Avg. Gain $\downarrow$ |
|---|---|---|---|---|---|---|---|
| | Original | + LS | Original | + LS | Original | + LS | |
| AugMix | 0.024 | 0.006 ($-0.018$) | 0.071 | 0.014 ($-0.057$) | 0.088 | 0.054 ($-0.034$) | **$-0.045$** |
| PixMix | 0.058 | 0.016 ($-0.042$) | 0.109 | 0.036 ($-0.073$) | 0.075 | 0.048 ($-0.027$) | **$-0.047$** |
| RandAug | 0.076 | 0.013 ($-0.063$) | 0.126 | 0.037 ($-0.089$) | 0.094 | 0.057 ($-0.037$) | **$-0.063$** |
| StyleAug | 0.095 | 0.014 ($-0.081$) | 0.169 | 0.039 ($-0.130$) | 0.072 | 0.047 ($-0.025$) | **$-0.079$** |
| CutOut | 0.085 | 0.013 ($-0.072$) | 0.135 | 0.039 ($-0.096$) | 0.094 | 0.056 ($-0.038$) | **$-0.069$** |

## 5.1 DATA AUGMENTATIONS AND LABEL SMOOTHING

**Data Augmentation Strategies:** Existing augmentation methods can be grouped into model-free, model-based, and policy-based algorithms, borrowing the taxonomy suggested by Xu et al. (2023).

*Model-free augmentations* can be divided into single-image and multiple-image methods. Single-image augmentations not only include elementary image manipulations such as translation, rotation, and color jittering but also includes masking methods such as CutOut (DeVries & Taylor, 2017) and Hide-and-Seek (Kumar Singh & Jae Lee, 2017). Multi-image augmentations exploit two or more images to construct augmented images, including the mixing of the two images, such as Mixup Zhang et al. (2018) and CutMix (Yun et al., 2019). Recently, it is empirically observed that a few augmentation methods including AugMix (Hendrycks et al., 2021b) and PixMix (Hendrycks et al., 2022) exhibit their superiority over others in terms of model robustness.

*Model-based augmentations* utilize pretrained models to generate augmented data. A number of methods that exploit generative models including CGAN (Mirza & Osindero, 2014), and their variants (Douzas & Bacao, 2018; Mariani et al., 2018; Ali-Gombe & Elyan, 2019; Yang & Zhou, 2021) aim to aid data imbalance problems. Other augmentations, including DeepAugment (Hendrycks et al., 2021a), ANT (Rusak et al., 2020), and StyleAug (Jackson et al., 2019), directly aim to enhance the classifier's robustness against common corruption, adversarial attacks, or domain shifts.

*Policy-based augmentations* focus on designing an automatic way to determine the optimal augmentation strategies by employing reinforcement learning or adversarial training. From a pioneering method called AutoAugment (Cubuk et al., 2019) that utilizes reinforcement learning for finding the best augmentation strategies, subsequent works such as Fast AA (Lim et al., 2019), Faster AA (Hataya et al., 2020), and RandAugment (Cubuk et al., 2020) aim to enhance both the efficiency of policy search and the model performance. Adversarial training-based augmentation strategies, including AdaTransform (Tang et al., 2019), Adversarial AA (Zhang et al., 2020), and AugMax (Wang et al., 2021) leverage adversarial perturbations which maximally disturbs samples to be misclassified into other labels, finally leading to improve model robustness against unseen domains.

Despite the effectiveness of augmentation methods in enhancing model performance, prior studies have focused more on their practical use rather than on understanding their theoretical impact on model robustness to data shifts. In our experiments, we carefully select representative methods from each group: **AugMix**, **PixMix**, **CutOut** from model-free methods, **StyleAug** from model-based method, **RandAugment** from policy-based methods.

**Label Smoothing:** Label smoothing (Szegedy et al., 2016) is a regularization strategy to enhance model robustness and generalization by softening training data targets. It reduces model overconfidence, especially in high-dimensional spaces, to prevent overfitting.

Empirical evidence supports label smoothing's benefits across multiple tasks and models. Müller et al. (2019) found that it improves deep neural networks' accuracy and robustness by reducing noise overfitting. Vaswani et al. (2017) incorporated it into the Transformer model, achieving the top performance in machine translation, underscoring its value in sequence-to-sequence models.

Label smoothing also enhances model calibration, notably reducing Expected Calibration Error (ECE). Studies, such as those by Guo et al. (2017) and Thulasidasan et al. (2019), highlight its

role in improving calibration and adversarial robustness, making models more reliable in critical applications. Mukhoti et al. (2020) demonstrated its significance on deep neural network calibration.

## 5.2 Augmentations, Label Smoothing and Model Robustness

**Augmentations and Model Robustness:** A number of previous works have tried to reveal the relationship between augmentation and model robustness. Zhao et al. (2020) and Volpi et al. (2018) have theoretically found out that adversarial perturbations in the latent space can simulate worst-case distributional shifts in the data. Rebuffi et al. (2021) have empirically found out that Mixup and CutOut with model weight averaging is shown to improve adversarial robustness. Najafi et al. (2019) and Alayrac et al. (2019) have theoretically shown that utilizing unlabeled data in training can improve adversarial robustness. Hendrycks et al. (2021a) have empirically found out that exploiting diverse augmentations together can improve model robustness against both adversarial and common data corruptions. Yin et al. (2019) interpret the augmentation-originated gains in model robustness by explaining that augmentations make deep models utilize both high and low frequency information of images so as to enhance model robustness against data corruptions.

Nonetheless, most of the prior interpretations of how augmentation contributes to model robustness are either confined to specific augmentations, adversarial robustness, or empirical analysis. None of the prior works generally explain how augmentations theoretically improve model robustness across various, along with the extensive empirical testings across augmentations and benchmarks.

**Label Smoothing and Model Robustness:** Shafahi et al. (2019) have empirically discovered that the benefits of adversarial training in boosting classifier robustness can also be achieved through simpler regularization techniques like label smoothing and logit squeezing. Similarly, Fu et al. (2020) have found that label smoothing not only results in logit squeezing but also enhances adversarial robustness. However, Goibert & Dohmatob (2019) have shown that while label smoothing may improve a model's defense against gradient-based adversarial attacks to some extent, this robustness is neither stable nor consistent. Additionally, Ren et al. (2022) have theoretically discovered that label smoothing promotes logit squeezing and smoothens the loss surface in the input space, which probably connects to the adversarial robustness.

The related works consistently show a clue of the benefits of label smoothing against adversarial robustness but do not formally prove the reduced empirical risks against adversarial samples.

**Augmentations with Label Smoothing and Model Robustness:** While augmentations with label smoothing have been empirically shown to potentially reduce calibration or adversarial error (Ren et al., 2022; Shafahi et al., 2019), there's a lack of research on their combined effect on model robustness against distributional shifts, including adversarial and common corruptions.

We fill this gap, examining how augmentations and label smoothing together can improve robustness against various distributional changes, highlighting the significance of our research in this area.

**Comparison with Fairness under Distribution Shift:** While Jiang et al. (2024) focuses on maintaining fairness under distribution shifts by highlighting the existence of model weight perturbations, our work provides a stricter formalization by demonstrating that input space augmentations with bounded $L_2$ norms can be transformed into equivalent parameter-space regions and vice versa. This duality allows us to rigorously establish how augmentations and label smoothing enhance model robustness against various distributional shifts.

## 6 Conclusion

This paper presents a theoretical framework that clarifies how data augmentations and label smoothing enhance model robustness through the lens of model flatness, generalization bounds, and adversarial robustness. With the theoretical claims, we provide extensive simulations verifying each claim via testing the various flatness metrics, corruption errors, domain generalization, and adversarial loss with various augmentations. Our findings aim to establish a general foundation for understanding the benefits of augmentations and label smoothing, which provide intuitions of future advancements in creating robust deep models capable of handling real-world data variability.

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

## A    SUPPLEMENTARY MATERIALS

This appendix contains additional material that could not be incorporated into the main paper due to page constraints, including detailed proofs of the theorems and experimental details. Section B provides detailed proofs of Theorems 1 to 5, Section C presents experimental details and additional results, and and Section D offers further high-level discussions on augmentations, model generalizations, and broader applicability of our findings.

## B    PROOFS ON THEOREMS

### B.1    PROOF ON THEOREM 1

**Theorem 1.** *(A closed-form of $\mathcal{R}_\mathcal{X}^\gamma$) Given $x \in \mathbb{R}^n$ and $\theta \in \mathbb{R}^{c \times n}$, for perturbation bounded with $L_2$-norm of $\gamma$ on parameter space, i.e., $\{\Delta \in \mathbb{R}^{c \times n} \mid \|\Delta\| \leq \gamma\}$, $\mathcal{R}_\mathcal{X}^\gamma$ is a $c$-dimensional rotated ellipsoid centered at $x$:*

$$\mathcal{R}_\mathcal{X}^\gamma = \{\delta \in \mathbb{R}^n \mid \delta^\top U^\top D U \delta \leq 1\}, \tag{6}$$

*where $U$ is from Singular Value Decomposition the model parameter $\theta$ with a form of $(c \times n)$ matrix: $\theta = U\Sigma V^\top$. Also, $D$ is the $(c \times c)$ diagonal matrix whose $i$-th diagonal element is $\sigma_i \|x\| \gamma$, where $\sigma_i$ is the $i$-th singular value of $\theta$.*

*Proof.* Let $f : x \mapsto \sigma(\theta x + b)$ represent a classifier with sigmoid activation $\sigma(\cdot)$. Given weights $\theta \in \mathbb{R}^{c \times n}, b \in \mathbb{R}^c$, input $x \in \mathbb{R}^n$, and parameter perturbation region $\|\Delta\| \leq \gamma$, we want to find the region $\mathcal{R}_\mathcal{X}^\gamma$ so that for all $\|\Delta\| \leq \gamma$, there exists $\delta \in \mathcal{R}_\mathcal{X}^\gamma$ *s.t.* $\sigma(\theta(x + \delta) + b) = \sigma((\theta + \Delta)x + b)$ and for all $\delta \in \mathcal{R}_\mathcal{X}^\gamma$, there exists $\|\Delta\| \leq \gamma$ *s.t.* $\sigma(\theta(x + \delta) + b) = \sigma((\theta + \Delta)x + b)$. In other words, we want to find the region $\mathcal{R}_\mathcal{X}^\gamma$ so that for every element $e_1$ in region $\{\Delta \in \mathbb{R}^{c \times n} \mid \|\Delta\| \leq \gamma\}$ there exists an element $e_2$ in region $\mathcal{R}_\mathcal{X}^\gamma$ satisfying the equation and vice versa.

Since $\sigma(\cdot) : \mathbb{R}^c \to (0,1)^c$ is a bijective function, $\sigma(\theta(x + \delta) + b) = \sigma((\theta + \Delta)x + b) \iff \theta(x + \delta) + b = (\theta + \Delta)x + b$. This equality can be reduced to $\theta\delta = \Delta x$.

We will first examine the range of $\Delta x$ in the output space, given $\|\Delta\| \leq \gamma$. $\Delta x$ can be written in several ways:

$$\Delta x = \begin{bmatrix} | & | & & | \\ v_1 & v_2 & \cdots & v_n \\ | & | & & | \end{bmatrix} \begin{bmatrix} x_1 \\ x_2 \\ \vdots \\ x_n \end{bmatrix} = \begin{bmatrix} v_{11} & v_{12} & \cdots & v_{1n} \\ v_{21} & v_{22} & \cdots & v_{2n} \\ \vdots & \vdots & & \vdots \\ v_{c1} & v_{c2} & \cdots & v_{cn} \end{bmatrix} \begin{bmatrix} x_1 \\ x_2 \\ \vdots \\ x_n \end{bmatrix}$$

$$= v_1 x_1 + v_2 x_2 + \cdots + v_n x_n = \begin{bmatrix} v_{11} \\ v_{21} \\ \cdots \\ v_{c1} \end{bmatrix} x_1 + \begin{bmatrix} v_{12} \\ v_{22} \\ \cdots \\ v_{c2} \end{bmatrix} x_2 + \cdots + \begin{bmatrix} v_{1n} \\ v_{2n} \\ \cdots \\ v_{cn} \end{bmatrix} x_n$$

, where $v_i$ is the $i$th column vector and $v_{ij}$ is an element in $i$th row, $j$th column of $\Delta$.

Next, we will rewrite $\|\Delta\| \leq \gamma$ as the following constraints:

$\|\Delta\| \leq \gamma$

$\iff \sum_{i=1}^{c} \sum_{j=1}^{n} v_{ij}^2 \leq \gamma^2$

$\iff \sum_{j=1}^{n} \|v_j\|^2 \leq \gamma_j^2$ subject to $\gamma_1^2 + \gamma_2^2 + \cdots + \gamma_n^2 = \gamma^2$.

When we reexamine the above formulas in $\mathbb{R}^c$, finding the range of $\Delta x$ can be regarded as finding the range of linear combination of column vectors in $\mathbb{R}^c$ such that each column vector $v_i$ is restricted to $\|v_i\| \leq \gamma_i$.

Given two vectors $u_1$ and $u_2$ s.t. $\|u_1\| \leq \gamma_1$ and $\|u_2\| \leq \gamma_2$, $\|u_1 + u_2\| \leq \gamma_1 + \gamma_2$. Trivially, for any $\alpha \in \mathbb{R}$, $\|\alpha \cdot u_1\| \leq |\alpha| \gamma_1$. That is, the range of linear combination $\Delta x = v_1 x_1 + v_2 x_2 + \cdots + v_n x_n$ is also a ball, i.e. $\|\Delta x\| \leq \sum_{i=1}^{n} |x_i| \gamma_i$ subject to $\sum_{i=1}^{n} \gamma_i^2 = \gamma^2$.

Finding the range of $\|\Delta x\|$ is now equivalent to finding the maximum radius of $\sum_{i=1}^{n} |x_i|\gamma_i$ with the constraint $\sum_{i=1}^{n} \gamma_i^2 = \gamma^2$. Using Lagrange multipliers method, let $r := [\gamma_1, \gamma_2, \cdots, \gamma_n]$, $f(r) := \sum_{i=1}^{n} |x_i|\gamma_i$, $g(r) := \sum_{i=1}^{n} \gamma_i^2 - \gamma^2$, and $L(r, \lambda) := f(r) - \lambda(g(r))$.

$$\frac{\partial L}{\partial \gamma_i} = |x_i| - 2\lambda\gamma_i = 0 \iff \gamma_i = \frac{|x_i|}{2\lambda}$$

Substituting the above equality to $g(r) = 0$,

$$\sum_{i=1}^{n} \frac{x_i^2}{4\lambda^2} - \gamma^2 = 0 \iff \lambda = \frac{\sqrt{\sum x_i^2}}{2\gamma}$$

$$\gamma_i = \frac{|x_i|}{2\lambda} = \frac{|x_i|\gamma}{\sqrt{\sum x_i^2}}$$

$$f(r) = \sum_{i=1}^{n} \frac{x_i^2 \gamma}{\sqrt{\sum x_i^2}} = \frac{\sum_{i=1}^{n} x_i^2}{\sqrt{\sum_{i=1}^{n} x_i^2}}\gamma = \|x\| \cdot \gamma$$

Therefore, $\|\Delta x\| \le \|x\|\gamma$.

We now consider the LHS of equation $\theta\delta = \Delta x$. Let $\theta = U\Sigma V^\top$ be the SVD Decomposition of $\theta \in \mathbb{R}^{c \times n}$. Multiplying $U^\top$ to both sides of the equation, $\Sigma V^\top \delta = U^\top \Delta x$. The inequality induced by $L_2$ norm, i.e. ball, does not change when we multiply any orthogonal matrix. Thus, $\|U^\top \Delta x\| \le \|x\|\gamma$.

Let $\delta' := V^\top \delta = [\delta'_1, \cdots, \delta'_n]^\top$.

$$\Sigma V^\top \delta = \Sigma \delta' = \begin{bmatrix} \sigma_1 & & & 0 & \cdots & 0 \\ & \ddots & & \vdots & & \vdots \\ & & \sigma_c & 0 & \cdots & 0 \end{bmatrix} \begin{bmatrix} \delta'_1 \\ \vdots \\ \delta'_c \\ \delta'_{c+1} \\ \vdots \\ \delta_n \end{bmatrix} = \begin{bmatrix} \sigma_1 \delta'_1 \\ \vdots \\ \sigma_c \delta'_c \end{bmatrix}$$

Since $\|U^\top \Delta x\| \le \|x\|\gamma$ and $\Sigma \delta' = U^\top \Delta x$, $\|\Sigma \delta'\| \le \|x\|\gamma$, i.e.

$$\sigma_1^2 \delta_1'^2 + \cdots + \sigma_c^2 \delta_c'^2 + 0 \cdot (\sigma_{c+1}^2 \delta_{c+1}'^2 + \cdots + \sigma_n^2 \delta_n'^2) \le \|x\|^2 \gamma^2$$

However since $0 \cdot (\sigma_{c+1}^2 \delta_{c+1}'^2 + \cdots + \sigma_n^2 \delta_n'^2) = 0$ holds for any $\delta$, i.e. the general solution to $\theta a = \theta b$ where $a \ne b$, we need not contain it in our perturbation region $\mathcal{R}_{\mathcal{X}}^\gamma$ which is induced by $\|\Delta\| \le \gamma$. Then, the above inequality represents a $c$-dim region bounded by an ellipsoid whose principal semi-axes have lengths $(\sigma_1 \|x\|\gamma)^{-1}, \cdots, (\sigma_c \|x\|\gamma)^{-1}$ with respect to $\delta' \in \mathbb{R}^n$. Subsequently, the region of interest $\mathcal{R}_{\mathcal{X}}^\gamma \subset \mathbb{R}^n$ is an rotated $c$-dim ellipsoid whose principal semi-axes have lengths $(\sigma_1 \|x\|\gamma)^{-1}, \cdots, (\sigma_c \|x\|\gamma)^{-1}$ with respect to $\delta \in \mathbb{R}^n$. $\qquad\square$

## B.2 PROOF ON THEOREM 2

Prior to delving into the proof of Theorem 2, we first establish the groundwork by proving the following Lemma:

**Lemma 1.** *Let $\mathcal{R}$ be the region of $x \in \mathbb{R}^n$ satisfying the inequality $x^\top A x \leq 1$, where $A$ is a non-zero positive semi-definite matrix having $\sigma_{max}$ as the maximum nonzero singular value. Let $\mathcal{R}'$ be the region of $x \in \mathbb{R}^n$ satisfying the ineqaulity $x^\top x \leq \sigma_{max}^{-1}$. Then, $\mathcal{R} \subseteq \mathcal{R}'$.*

We handle two cases where $rank(A) = m$ and $rank(A) < m$. (We use $[n]$ to represent set $\{1, 2, \cdots, n\}$ henceforth.)

**Case $rank(A) = m$:**

Using SVD Decomposition, $A = U\Sigma U^\top$, where $\Sigma = \begin{bmatrix} \sigma_1 & & \\ & \ddots & \\ & & \sigma_n \end{bmatrix}$

$x^\top A x = x^\top U\Sigma U^\top x = x'^\top \Sigma x' \leq 1$, where $x' := U^\top x$

Let $x'$ be represented as $x' = [x'_1, \cdots, x'_n]$.

The constraint induced by $\mathcal{R}$ can be rewritten as:

$$x'^\top \Sigma x' = \sigma_1 x_1'^2 + \cdots + \sigma_n x_n'^2 \leq 1, \text{ where } \Sigma = U^\top A U$$

Let $x \in \mathbb{R}^n$ be some vector satisfyig $x^\top x \leq \sigma_{max}$. Since $U$ is an orthogonal matrix and $x^\top x \leq \sigma_{max}^{-1}$ is an equidistant ball that is invariant under rotations and reflections, the constraint induced by $\mathcal{R}'$ can be rewritten as $x'^\top x' \leq \sigma_{max}^{-1}$, where $x' = U^\top x$.

To prove $x \in \mathcal{R}'$ implies $x \in \mathcal{R}$, we will show $x'^\top x' \leq \sigma_{max}^{-1}$ implies $x'^\top \Sigma x' \leq 1$.

$$x'^\top x' \leq \sigma_{max}^{-1} \iff \sigma_{max} x'^\top x' \leq 1$$

Let $\epsilon_i := \sigma_{max} - \sigma_i$. Then, $\forall i \in [n], \epsilon_i \geq 0$.

$$\sigma_{max} x'^\top x' - \sum_{i=1}^n \epsilon(x'_i)^2 \leq 1 - \sum_{i=1}^n \epsilon_i(x'_i)^2 \qquad (\because \sigma_{max} x'^\top x' \leq 1)$$
$$\leq 1 \qquad (\because \forall i \in [n], \epsilon_i(x'_i)^2 \geq 0)$$

**Case $rank(A) < m$:**

Let $rank(A) = k < m$. $A$ can be represented as $U\Sigma U^\top$ using SVD decomposition, where $\Sigma$ is a diagonal matrix whose first $k$ elements are non-zero singular values $\sigma_1, \cdots, \sigma_k$.

$$x^\top A x = x^\top U\Sigma U^\top x = x'\Sigma x' \leq 1, \text{ where } \Sigma = U^\top A U \text{ and } x' := U^\top x$$

Let $x'$ be represented as $[x'_1, \cdots, x'_n]$. The constrained induced by $\mathcal{R}$ can be rewritten as:

$$x'^\top \Sigma x' = \sigma_1 x_1'^2 + \cdots + \sigma_k x_k'^2 \leq 1$$

Let $x \in \mathbb{R}^n$ be any vector satisfying $x^\top x \leq \sigma_{max}^{-1}$. Since ball is equidistant, $x^\top x \leq \sigma_{max}^{-1} \iff x''^\top x' \leq \sigma_{max}^{-1}$, where $x' = U^\top x$.

To prove $x \in \mathcal{R}'$ implies $x \in \mathcal{R}$, we will show $x'^\top x' \leq \sigma_{max}^{-1}$ implies $x'\Sigma x' \leq 1$.

$$x'^\top x' \leq \sigma_{max}^{-1} \iff \sigma_{max} x'^\top x' \leq 1 \iff \sum_{i=1}^n \sigma_{max}(x'_i)^2 \leq 1$$

Let $\epsilon_i := \sigma_{max} - \sigma_i$. Then, $\forall i \in [n], \epsilon_i \geq 0$.

$$\sum_{i=1}^{k}(\sigma_{max} - \epsilon_i)x_i'^2 \leq \sigma_{max}x^\top x - \sum_{i=1}^{k}\epsilon_i(x_i')^2 \qquad \left(\because \sum_{i=k+1}^{n}\sigma_{max}x_i^2 \geq 0\right)$$

$$\leq 1 - \sum_{i=1}^{k}\epsilon_i(x_i')^2 \qquad \left(\because \sigma_{max}x'^\top x' \leq 1\right)$$

$$\leq 1 \qquad \left(\because \forall i \in [k], \epsilon_i(x_i')^2 \geq 0\right)$$

Since $\sum_{i=1}^{k}(\sigma_{max} - \epsilon_i)x_i'^2 = x'^\top \Sigma x'$, $x'^\top \Sigma x' \leq 1$.

With Lemma 1 established, we now turn our attention to the proof of Theorem 2.

**Theorem 2.** *(A subset of $\mathcal{R}_\Theta^\gamma$, i.e., $\mathcal{R}_{\Theta,\mathrm{sub}}^\gamma$) Given dataset $\mathcal{D} = \{x_i\}_{i=1}^{N}$ and $\theta \in \mathbb{R}^{c \times n}$, for pertur-bation bounded with $L_2$-norm of $\gamma$ on input space, i.e., $\{\delta \in \mathbb{R}^n \mid \|\delta\| \leq \gamma\}$, $\mathcal{R}_{\Theta,\mathrm{sub}}^\gamma$ is the subset of $\mathcal{R}_\Theta^\gamma$, i.e., $\mathcal{R}_{\Theta,\mathrm{sub}}^\gamma \subseteq \mathcal{R}_\Theta^\gamma$:*

$$\mathcal{R}_{\Theta,\mathrm{sub}}^\gamma = \{\Delta \in \mathbb{R}^{c \times n} \mid \|\Delta\| \leq \gamma^2 \sigma_{\min}^2/\|x_{\max}\|^2\}, \tag{7}$$

*where $\sigma_{\min}^2 := \min_i \sigma_i^2$ is the minimum singular value of $\theta$ and $x_{\max} := \max_{x \in \mathcal{D}} \|x\|$.*

*Proof.* Given weights $\theta \in \mathbb{R}^{c \times n}$, input $x \in \mathbb{R}^n$, and parameter perturbation region $\|\delta\| \leq \gamma$, we want to find the region $\mathcal{R}_\mathcal{X}^\gamma$ so that for all $\|\delta\| \leq \gamma$, there exists $\Delta \in \mathcal{R}_\mathcal{X}^\gamma$ s.t. $\theta\delta = \Delta x$ and for all $\Delta \in \mathcal{R}_\mathcal{X}^\gamma$, there exists $\|\delta\| \leq \gamma$ s.t. $\theta\delta = \Delta x$ as in Theorem 1's proof ( B.1.)

Using SVD decomposition, $\theta = U\Sigma V^\top$, where $\Sigma$ is a diagonal matrix with entries $\sigma_1, \cdots, \sigma_n$.

$\theta\delta = U\Sigma V^\top \delta = U\Sigma\delta'$, where $\delta' := V^\top \delta$. Since rotating or reflecting does not change the region of a ball, $\|\delta\| \leq \gamma$ gives $\|\delta'\| \leq \gamma$, i.e. $\delta_1'^2 + \cdots \delta_n'^2 \leq \gamma^2$.

Let $\delta'' := [\delta_1'', \cdots, \delta_c''] = \Sigma\delta' = [\sigma_1\delta_1', \cdots, \sigma_c\delta_c']$. $\forall i \in [c], \sigma_i^{-1}\delta_i'' = \delta_i'$. Then,

$$\frac{\delta_1''^2}{\sigma_1^2} + \cdots + \frac{\delta_c''^2}{\sigma_c^2} \leq \gamma^2 - \left(\delta_{c+1}'^2 + \cdots \delta_n'^2\right) \tag{8}$$

The maximum value of RHS in the above equation is $\gamma^2$, when $\left(\delta_{c+1}'^2 + \cdots \delta_n'^2\right) = 0$. This indicates that $\delta''$ resides within an ellipsoid with with principle semi-axes of lengths $\lambda_i := \sigma_i\gamma, i \in [c]$. Thus, $U\delta'' = \theta\delta$ is a region bounded by an rotated ellipsoid.

Now, we will examine the region $\mathcal{R}_\mathcal{X}^\gamma$ such that $\Delta x$ ($\Delta \in \mathcal{R}_\mathcal{X}^\gamma$) forms a rotated ellipsoid with principle semi-axes of lengths $\lambda_i$. Unlike the case of converting parameter space's perturbation region to input space, $\mathcal{R}_\mathcal{X}^\gamma$ need not be in a form of ellipsoid. Instead, we provide a subset $\mathcal{R}_{\Delta,sub}^\gamma$ of $\mathcal{R}_\mathcal{X}^\gamma$ in the form of a ball such that $\mathcal{R}_{\Delta,sub}^\gamma \subseteq \mathcal{R}_\mathcal{X}^\gamma$.

Let $\theta$ be decomposed into $U\Sigma V^\top$ using SVD decomposition. For now, we will consider the special case of $\theta$ where $U = I$, i.e. the region of $\theta\delta$ is bounded by an ellipsoid aligned with standard basis. Afterwards, we will consider the general case of $\theta$, i.e. the region of $\theta\delta$ is bounded by a rotated ellipsoid.

Let $d_{ij}$ denote the $i$th row, $j$th column element of $\Delta \in \mathbb{R}^{c \times n}$ and $x_i$ the $i$th element of $x \in \mathbb{R}^n$. Since the range of $\Delta x$ is an ellipsoid, $\Delta x$ must satisfy the ellipsoid inequality

$$\frac{(x_1d_{11} + x_2d_{12} + \cdots + x_nd_{1n})^2}{\lambda_1^2} + \cdots + \frac{(x_1d_{c1} + x_2d_{c2} + \cdots + x_nd_{cn})^2}{\lambda_c^2} \leq 1$$

Let $r_i$ denote the $i$th row vector of $\Delta$, and let $X$ denote $xx^\top$. The above inequality can be rewritten as:

$$\frac{r_1^\top X r_1}{\lambda_1^2} + \frac{r_2^\top X r_2}{\lambda_2^2} + \cdots + \frac{r_c^\top X r_c}{\lambda_c^2} \leq 1 \tag{9}$$

Since we are interested in finding the region of $\Delta$ in $\mathbb{R}^{c \times n}$ space, we may think of it as a vector $d = [r_1^\top, r_2^\top, \cdots, r_c^\top]$ in $\mathbb{R}^{(c \times n)} = \mathbb{R}^m$ rather than as a matrix. Then, inequation 2 can be rewritten as:

$$d^\top X_\lambda d \leq 1, \text{ where } X_\lambda := \begin{bmatrix} X/\lambda_1^2 & & & \\ & X/\lambda_2^2 & & \\ & & \cdots & \\ & & & X/\lambda_c^2 \end{bmatrix} \in \mathbb{R}^{m^2}$$

One property of $X_\lambda$ is that it is a rank $c$ matrix with singular values $\|x\|^2/\lambda_1^2, \cdots, \|x\|^2/\lambda_c^2$, regarding that $X/\lambda_i^2$ is a rank 1 matrix with singular value $\|x\|^2/\lambda_i^2$. Another property is that $X_\lambda$ is a positive-semidefinite matrix ($\because \forall i \in [c], \|x\|^2/\lambda_i^2 \geq 0$.)

When we think of a single input $x$, the area of $d$ satisfying $d^\top X_\lambda d \leq 1$ is not bounded. However, when we consider the constraint over multiple values of input datapoints $\{x_1, x_2, \cdots, x_N\}(N \gg n)$ that spans $\mathbb{R}^n$, the area becomes bounded. One justification of the multiple constraints is that when we consider $x$ a uniform random variable over the input datapoints, the region of $d$ that satisfies all the possible constraint is $\cap_{i=1}^N d^\top X_\lambda^{(i)} d \leq 1$, where $X_\lambda^{(i)}$ denotes $X_\lambda$ for $x = x_i$. Another justification is that when we reach a local plateau in training parameter $\theta$, there is little or no change in the value of $\theta$.

Now, we will rewrite Theorem 2 as the following statement:

**Theorem 2.** *Given a set of datapoints $\mathcal{D} = \{x_i\}_{i=1}^N$ ($x_i \in \mathbb{R}^n/\{0\}, i \in [N]$), let $\mathcal{R}_\Theta^\gamma$ be the region of $d \in \mathbb{R}^m$ satisfying the inequality $d^\top X_\lambda d \leq 1$ for all $x \in \mathcal{D}$. Let $\mathcal{R}_{\Theta,sub}^\gamma$ be the region of $d \in \mathbb{R}^m$ satisfying $d^\top d \leq (\|x_{max}\|^2/\lambda_{min}^2)^{-1}$, where $x_{max} := \arg\max_{x_i} \|x_i\|$ and $\lambda_{min} := \min\{\lambda_1, \cdots, \lambda_c\}$. $\mathcal{R}_{\Theta,sub}^\gamma \subseteq \mathcal{R}_\Theta^\gamma$.*

Remark that $X_\lambda^{(i)} = \begin{bmatrix} x_i^\top x_i/\lambda_1^2 & & & \\ & x_i^\top x_i/\lambda_2^2 & & \\ & & \cdots & \\ & & & x_i^\top x_i/\lambda_c^2 \end{bmatrix}$. $X_\lambda^{(i)}$ is a rank $c$ matrix with singular values $\|x_i\|^2/\lambda_1^2, \cdots, \|x_i\|^2/\lambda_c^2$.

Let $\mathcal{R}_i$ denote the region of $d \in \mathbb{R}^n$ satisfying $d^\top X_\lambda^{(i)} d \leq 1$, and let $\mathcal{R}_i'$ denote the region $d^\top d \leq \left(\frac{\|x_i\|^2}{\lambda_{min}^2}\right)^{-1} \cdot \frac{\|x_i\|^2}{\lambda_{min}^2}$ being the largest singular value of $X_\lambda^{(i)}$, $\mathcal{R}_i' \subseteq \mathcal{R}_i$ by Lemma 1. Since this holds for all $i \in [N]$, $\bigcap_{i=1}^N \mathcal{R}_i' \subseteq \bigcap_{i=1}^N \mathcal{R}_i$. $\bigcap_{i=1}^N \mathcal{R}_i = \mathcal{R}_\Theta^\gamma$, and $\bigcap_{i=1}^N \mathcal{R}_i' = \mathcal{R}_{\Theta,sub}^\gamma$ is a ball with smallest radius, i.e. $d^\top d \leq \left(\frac{\|x_{max}\|^2}{\lambda_{min}^2}\right)^{-1} = \gamma^2 \sigma_{min}^2/\|x_{max}\|^2$.

We have so far addressed the case where $U = I$ for $\theta = U\Sigma V^\top$ in the equation $\theta\delta = \Delta x$. Now, let us consider the general case of full rank matrix $\theta$.

$\Delta \in \mathbb{R}^{c \times n}$ can be represented as either column vectors $[v_1, v_2, \cdots, v_n]$ or row vectors $[r_1, r_2, \cdots, r_c]^\top$. The equation $\theta\delta = \Delta x$ can be rewritten as:

$$\Sigma V^\top \delta = U^\top \Delta x = U^\top [v_1, v_2, \cdots, v_n]x = [U^\top v_1, U^\top v_2, \cdots, U^\top v_n]x$$

Let $\Delta' := U^\top \Delta = [v_1', v_2', \cdots, v_n'] = [r_1', r_2', \cdots, r_c']^\top$, and let $d'$ be the flattened vector representation $[r_1'^\top, r_2'^\top, \cdots, r_c'^\top]$ of $\Delta'$. Then, finding $\mathcal{R}_\Theta^\gamma$ is equivalent to finding the region of $\Delta'$ satisfying $d'^\top X_\lambda d' \leq 1$ and multiplying $U$ to $\Delta'$.

The relationship between $\Delta'$ and $\Delta$ can be expressed as:

$$
U_{diag}\begin{bmatrix} v'_1 \\ v'_2 \\ \vdots \\ v'_n \end{bmatrix} = \begin{bmatrix} v_1 \\ v_2 \\ \vdots \\ v_n \end{bmatrix}, \text{ where } U_{diag} := \begin{bmatrix} U & & & \\ & U & & \\ & & \ddots & \\ & & & U \end{bmatrix} \in \mathbb{R}^{m^2}
$$

$U_{diag}$ is an orthogonal matrix since $U$ is an orthogonal matrix. Furthermore, any permutation $\pi$ that permutes the row vectors of $U_{diag}$ also results in another orthogonal matrix $U_{diag}^{\pi}$. Then for some $\pi$, $U_{diag}^{\pi}[r_1'^{\top}, r_2'^{\top}, \cdots, r_c'^{\top}]^{\top} = [r_1^{\top}, r_2^{\top}, \cdots, r_c^{\top}]^{\top}$, i.e. $U_{diag}^{\pi} d' = d$. Since the region of a ball is not affected by rotations or reflections, the subset obtained is not affected. In other words,

$$
\mathcal{R}_{\mathcal{X}}^{\gamma} = \{d \in \mathbb{R}^{c \times n} \mid \forall i \in [n], d^{\top} X_{\lambda}^{(i)} d \le 1\}
$$

$$
\mathcal{R}_{\Delta,sub}^{\gamma} = \{d \in \mathbb{R}^{c \times n} \mid d^{\top} d \le \gamma^2 \sigma_{\min}^2 / \|x_{max}\|^2\}
$$

satisfies $\mathcal{R}_{\Delta,sub}^{\gamma} \subseteq \mathcal{R}_{\mathcal{X}}^{\gamma}$. □

### B.3 Proof on Theorem 3

**Theorem 3.** *(Flatness of $\tilde{\theta}^*$) Let $\theta^* \in \Theta^*$ and $\tilde{\theta}^* \in \tilde{\Theta}^*$ be $b^*$ and $\tilde{b}^*$-flat minima, respectively. The following inequality holds:*

$$
\min_{\theta^* \in \Theta^*} b^* \le \min_{\tilde{\theta}^* \in \tilde{\Theta}^*} \tilde{b}^*. \tag{10}
$$

*Proof.* We consider $f(\cdot; \theta)$ to be a universal approximator for continuous functions from the input space $\mathbb{R}^n$ to the output space $\mathbb{R}^c$. For the sake of simplicity, we will omit notations $\theta^* \in \Theta^*$ and $\tilde{\theta}^* \in \tilde{\Theta}^*$ in $\min_{\theta^* \in \Theta^*}$ and $\min_{\tilde{\theta}^* \in \tilde{\Theta}^*}$ henceforth. We prove $\min b^* = 0 < (\gamma \sigma_{min})^2 / \|x_{max}\|^2 \le \min \tilde{b}^*$.

Let $\theta_0^*$ be an optimal parameter of a model that satisfies $\forall (x, y) \in \mathcal{D}, \mathcal{L}(f(x; \theta_0^*), y) = 0$ and $\forall (x, y) \in \mathbb{R}^n \times \mathbb{R}^c / \mathcal{D}, \mathcal{L}(f(x; \theta_0^*), y) > 0$.

Let $b_0^*$ denote the $b$-value of the $b$-flat minima for $\theta_0^*$. Suppose $b_0^* = h > 0$. By the definition of $b$-flat minima, $\forall \|\Delta\| \le h, \forall (x, y) \in \mathcal{D}, \mathcal{L}(f(x; \theta_0^* + \Delta), y) = \mathcal{L}(f(x; \theta_0^*), y))$. By Theorem 1, $\exists \delta \in \mathcal{R}_{\mathcal{X}}^{\gamma}$ such that $\mathcal{L}(f(x + \delta; \theta_0^*), y) = \mathcal{L}(f(x; \theta_0^*), y) = 0$ and $\|\delta\| > 0$ for some $(x, y) \in \mathcal{D}$. Nonetheless, $\mathcal{L}(f(x + \delta; \theta_0^*), y) > 0$ by the definition of $\theta_0^*$. $\Rightarrow \Leftarrow$

This implies $\min b^* \le \min b_0^* \le 0, i.e. \min b^* = 0$.

We now show $(\sigma_{min} \gamma)^2 / \|x_{max}\|^2 \le \min \tilde{b}^*$. Let $\mathcal{D}_{\mathcal{A}} = \{(\tilde{x}, y) \mid \tilde{x} := \mathcal{A}(x), (x, y) \sim \mathcal{D}\}$ represent an augmented dataset without label smoothing. Let $\tilde{\theta}_x^*$ be some optimal parameter that satisfies $\forall (\tilde{x}, y) \in \mathcal{D}_{\mathcal{A}}, \mathcal{L}(f(\tilde{x}; \tilde{\theta}_x^*), y)) = 0$, and $\tilde{b}_x^*$ be the $b$ value for the $b$-flat minima w.r.t. $\mathcal{D}$. We show that $(\sigma_{min} \gamma)^2 / \|x_{max}\|^2 \le \min \tilde{b}_x^* = \min \tilde{b}^*$.

By definition, $\gamma$ represents the maximum $L_2$ distance that augmentation has non-zero probabilities around the original input. Using Theorem 2 and the definition of $\gamma$ and $\tilde{b}_x^*$, we have $\gamma^2 \sigma_{min}^2 / \|x_{max}\|^2 \le \tilde{b}_x^*$.

By definition of $\tilde{\theta}^*$, for all $(x, y) \in \mathcal{D}$,

$$
\begin{aligned}
\mathcal{L}(f(\tilde{x}; \tilde{\theta}^*), \tilde{y}) &= \tilde{y}^T \log f(\tilde{x}; \tilde{\theta}^*) \\
&= (1 - \epsilon) y^T \log f(\tilde{x}; \tilde{\theta}^*) + \epsilon \cdot \mathbf{1}^T \log f(\tilde{x}; \tilde{\theta}^*) = 0.
\end{aligned}
$$

$$\mathcal{L}(f(\tilde{x}; \tilde{\theta}^*), y) = y^T \log f(\tilde{x}; \tilde{\theta}^*)$$
$$= -\frac{\epsilon}{1-\epsilon} \big(\log(1 - \epsilon + \frac{\epsilon}{c}) + (c-1) \cdot \log\frac{\epsilon}{c}\big) = \alpha.$$

, where $\alpha > 0$ is some constant ($\because \epsilon$ and $c$ are constants.) This implies $\mathcal{L}(f(\tilde{x}; \tilde{\theta}^*), y)$ is constant for all $(x, y) \in \mathcal{D}$, i.e. $\tilde{b}^* = \tilde{b}_x^*$ by the definition of $b$-flat minima.

$\square$

### B.4 PROOF ON THEOREM 4

**Theorem 4.** *(Generalization bound) Given $M$ covering sets $\{\Theta_k\}_{k=1}^M$ of parameter space $\Theta$ with $\Theta = \bigcup_{k=1}^M \Theta_k$ and $diam(\Theta) = \sup_{\theta, \theta' \in \Theta} ||\theta - \theta'||_2$, where $M = \left\lceil \frac{diam(\Theta)}{\gamma} \right\rceil^d$ ($d$ is the dimension of $\Theta$ and $\gamma$ is the value satisfying the condition of Assumption 1), and VC dimension $v_k$ for each $\Theta_k$, the following inequality holds with probability at least $1 - \delta$:*

$$\mathcal{E}_{\mathcal{T}}(\theta) < \hat{\mathcal{E}}_{\tilde{\mathcal{D}}}(\theta) + \frac{1}{2}\boldsymbol{Div}(\mathcal{D}, \mathcal{T}) + \log c + \max_{k \in [1,M]} \left[ \sqrt{\frac{v_k \log(N/v_k)}{N}} + \frac{\log(M/\delta)}{N} \right],$$

*where $\boldsymbol{Div}(\mathcal{D}, \mathcal{T}) = 2 \sup_A |\mathbb{P}_{\mathcal{D}}(A) - \mathbb{P}_{\mathcal{T}}(A)|$ measures the maximal discrepancy between distributions $\mathcal{D}$ and $\mathcal{T}$, and $N$ is the number of samples drawn from $\tilde{\mathcal{D}}$.*

*Proof.* We will use $\mathcal{D}_{\mathcal{A}}$ and $\tilde{\mathcal{D}}$ to represent augmented data distributions without and with label smoothings, respectively. Analogous to robust empirical risk introduced by Cha et al. (2021), we define $\hat{\mathcal{E}}_{\mathcal{D}}^{\gamma}(\theta) := \max_{\|\delta\| \leq \gamma} \frac{1}{N} \sum_{i=1}^N \left[ L(f(x_i + \delta, \theta), y_i) \right]$ with respect to $\mathcal{D}$ as the robust empirical risk defined on the *input space*.

We now list the following prepositions necessary for proving Theorem 6:

**Preposition 1.** $\hat{\mathcal{E}}_{\tilde{\mathcal{D}}}(\tilde{\theta}^*) = 0$, and $\hat{\mathcal{E}}_{\mathcal{D}_{\mathcal{A}}}(\tilde{\theta}^*) = \hat{\mathcal{E}}_{\mathcal{D}}^{\gamma}(\tilde{\theta}^*) = -\log(\epsilon - c + \epsilon/c)$.

*Proof.* $\hat{\mathcal{E}}_{\tilde{\mathcal{D}}}(\tilde{\theta}^*) = 0$ by the definition of $\tilde{\theta}^*$. $\hat{\mathcal{E}}_{\mathcal{D}_{\mathcal{A}}}(\tilde{\theta}^*) = -1 \cdot \log(\epsilon - c + \epsilon/c) - (c-1) \cdot 0 \cdot \log(\epsilon/c)$ $= -\log(\epsilon - c + \epsilon/c)$ by the definition of $\tilde{\theta}^*$. $\hat{\mathcal{E}}_{\mathcal{D}}^{\gamma}(\tilde{\theta}^*) = \hat{\mathcal{E}}_{\mathcal{D}_{\mathcal{A}}}(\tilde{\theta}^*)$ by the definition of $\gamma$.

Let $\gamma' := \gamma \cdot \lambda_{min}/\|x_{max}\|$, where $\gamma$ is the maximal nonzero region around the original image bounded by $L_2$ norm of size $\gamma$. We additionally utilize the following prepositions of Cha et al. (2021) to prove Theorem 6.

**Preposition 2.** For all $\theta \in \Theta$, $|\mathcal{E}_{\mathcal{D}}(\theta) - \mathcal{E}_{\mathcal{T}}(\theta)| \leq \frac{1}{2}\boldsymbol{Div}(\mathcal{D}, \mathcal{T})$

**Preposition 3.** For all $\theta \in \Theta$, $\mathcal{E}_{\mathcal{D}}(\theta) \leq \hat{\mathcal{E}}_{\mathcal{D}}^{\gamma'} + \max_{k \in [1,M]} \left[ \sqrt{\frac{v_k \log(N/v_k)}{N}} + \frac{\log(M/\delta)}{N} \right]$.

For the sake of readability, we will use $\mathcal{B}(M, N, \delta)$ interchangeably with $\max_{k \in [1,M]}$ $\left[ \sqrt{\frac{v_k \log(N/v_k)}{N}} + \frac{\log(M/\delta)}{N} \right]$. We are now ready to prove Theorem 6.

$$\mathcal{E}_T(\tilde{\theta}^*) \leq \mathcal{E}_{\mathcal{D}}(\tilde{\theta}^*) + \frac{1}{2}\mathbf{Div}(\mathcal{D}, \mathcal{T}) \qquad\qquad \text{Prep. 2}$$

$$\leq \hat{\mathcal{E}}_{\mathcal{D}}^{\gamma'}(\tilde{\theta}^*) + \frac{1}{2}\mathbf{Div}(\mathcal{D}, \mathcal{T}) + \mathcal{B}(M, N, \delta) \qquad\qquad \text{Prep. 3}$$

$$= \hat{\mathcal{E}}_{\mathcal{D}}^{\gamma}(\tilde{\theta}^*) + \frac{1}{2}\mathbf{Div}(\mathcal{D}, \mathcal{T}) + \mathcal{B}(M, N, \delta) \qquad\qquad \text{Def. } \gamma', \text{Thm. 2}$$

$$= \hat{\mathcal{E}}_{\mathcal{D}_{\mathcal{A}}}(\tilde{\theta}^*) + \frac{1}{2}\mathbf{Div}(\mathcal{D}, \mathcal{T}) + \mathcal{B}(M, N, \delta) \qquad\qquad \text{Prep. 1}$$

$$= \hat{\mathcal{E}}_{\tilde{\mathcal{D}}}(\tilde{\theta}^*) - \log\left(\epsilon - c + \frac{\epsilon}{c}\right) + \frac{1}{2}\mathbf{Div}(\mathcal{D}, \mathcal{T}) + \mathcal{B}(M, N, \delta) \qquad \text{Prep. 1}$$

$$< \hat{\mathcal{E}}_{\tilde{\mathcal{D}}}(\tilde{\theta}^*) + \log c + \frac{1}{2}\mathbf{Div}(\mathcal{D}, \mathcal{T}) + \mathcal{B}(M, N, \delta)$$

, where the last inequality holds since $-\log\left(\epsilon - c + \epsilon/c\right) < \log c$ for all $c > 1$ and $\epsilon \in (0, 1)$.

$\square$

## B.5 PROOF ON THEOREM 5

**Theorem 5.** *(Adversarial robustness) For the dataset of adversarial samples $\mathcal{D}_{\mathrm{adv}}$, the models parameterized by $\tilde{\theta}_{LS}^*$ and $\theta^*$ satisfy the following inequality:*

$$\hat{\mathcal{E}}_{\mathcal{D}_{\mathrm{adv}}}(\tilde{\theta}_{LS}^*) < \hat{\mathcal{E}}_{\mathcal{D}_{\mathrm{adv}}}(\tilde{\theta}^*). \tag{10}$$

Prior to delving into the proof, we introduce an assumption on the adversarial dataset:

**Assumption 2.** *There exists a datapoint $(x, y) \in \mathcal{D}$ and its adversarial counterpart $(x_{adv}, y) \in \mathcal{D}_{adv}$ such that:*

1. *Positive Augmentation Probability:*
$$\mathcal{P}_{\mathcal{A}}(x_{adv}|x) > 0.$$

2. *Exclusive Generation:*
$$\text{For all } (x', y') \in \mathcal{D}\backslash\{(x, y)\}, \ \mathcal{P}_{\mathcal{A}}(x_{adv}|x') = 0.$$

*Proof.* Let $(x_{\mathrm{adv},\kappa}, y_\kappa)$ denote a hard negative example satisfying **Assumption 2**.

Suppose there exists $\tilde{\theta}_{LS}^*$ and $\tilde{\theta}^*$ such that $\hat{\mathcal{E}}_{\mathcal{D}_{\mathrm{adv}}}(\tilde{\theta}_{LS}^*) \geq \hat{\mathcal{E}}_{\mathcal{D}_{\mathrm{adv}}}(\tilde{\theta}^*)$.

$$\hat{\mathcal{E}}_{\mathcal{D}_{\mathrm{adv}}}(\tilde{\theta}_{LS}^*) = -\frac{1}{|\mathcal{D}_{\mathrm{adv}}|} \sum_{(x_{\mathrm{adv}}, y) \in \mathcal{D}_{\mathrm{adv}}} y^\top \log f(x_{\mathrm{adv}}; \tilde{\theta}_{LS}^*)$$

$$= -\frac{1}{|\mathcal{D}_{\mathrm{adv}}|} \sum_{i \in [n]} \log h_i =: l_\epsilon$$

, where for all $i \in [n], 0 < h_i < 1$.

Let $\mathcal{D}_{\mathrm{adv}\backslash\kappa} := \mathcal{D}_{\mathrm{adv}}\backslash\{(x_{\mathrm{adv},\kappa}, y_\kappa)\}$.

$$\hat{\mathcal{E}}_{\mathcal{D}_{\mathrm{adv}}}(\tilde{\theta}^*) = -\frac{1}{|\mathcal{D}_{\mathrm{adv}}|} \sum_{(x_{\mathrm{adv}}, y) \in \mathcal{D}_{\mathrm{adv}}} y^\top \log f(x_{\mathrm{adv}}; \tilde{\theta}^*)$$

$$= -\frac{1}{|\mathcal{D}_{\mathrm{adv}} - 1|} \sum_{(x_{\mathrm{adv}}, y) \in \mathcal{D}_{\mathrm{adv}\backslash\kappa}} y^\top \log f(x_{\mathrm{adv}}; \tilde{\theta}^*) - y_\kappa^\top \log f(x_{\mathrm{adv},\kappa}; \tilde{\theta}^*)$$

Let $l := -\frac{1}{|\mathcal{D}_{\mathrm{adv}} - 1|} \sum_{(x_{\mathrm{adv}}, y) \in D_{\mathrm{adv} \setminus \kappa}} y^\top \log f(x_{\mathrm{adv}}, \tilde{\theta}^*)$.

We want to prove $l - y_\kappa^\top \log f(x_{\mathrm{adv},\kappa}; \tilde{\theta}^*) \le l_\epsilon$.

**Case $l > l_\epsilon$:**

Since $-y_\kappa^\top \log f(x_{\mathrm{adv},\kappa}; \tilde{\theta}^*) \ge 0$, $l - y_\kappa^\top \log f(x_{\mathrm{adv},\kappa}; \tilde{\theta}^*) \ge l > l_\epsilon$. $\Rightarrow\Leftarrow$

**Case $l = l_\epsilon$:**

$l - y_\kappa^\top \log f(x_{\mathrm{adv},\kappa}; \tilde{\theta}^*) \le l_\epsilon$, and $-y_\kappa^\top \log f(x_{\mathrm{adv},\kappa}; \tilde{\theta}^*) \le l_\epsilon - l = 0$.

Since $0 \le -y_\kappa^\top \log f(x_{\mathrm{adv},\kappa}; \tilde{\theta}^*)$, we have $-y_\kappa^\top \log f(x_{\mathrm{adv},\kappa}; \tilde{\theta}^*) = 0$. This implies $f(x_{\mathrm{adv},\kappa}; \tilde{\theta}^*) = y_\kappa$, which violates the definition of adversarial example $f(x_{\mathrm{adv},\kappa}; \tilde{\theta}^*) \ne y_\kappa$. $\Rightarrow\Leftarrow$

**Case $l < l_\epsilon$:**

$l - y_\kappa^\top \log f(x_{\mathrm{adv},\kappa}; \tilde{\theta}^*) \le l_\epsilon$, and $-y_\kappa^\top \log f(x_{\mathrm{adv},\kappa}; \tilde{\theta}^*) \le l_\epsilon - l$.

Since $0 \le -y_\kappa^\top \log f(x_{\mathrm{adv},\kappa}, \tilde{\theta}^*)$, $0 \le -y_\kappa^\top \log f(x_{\mathrm{adv},\kappa}; \tilde{\theta}^*) \le l_\epsilon - l$.

Let $i$ be the ground-truth index of $y_\kappa$ w.l.o.g. Then, $-y_\kappa^\top \log f(x_{\mathrm{adv},\kappa}; \tilde{\theta}^*) = -\log f(x_{\mathrm{adv},\kappa}; \tilde{\theta}^*)_i$.

$0 \le \log f(x_{\mathrm{adv},\kappa}; \tilde{\theta}^*)_i \le l_\epsilon - l$, and $1 \ge f(x_{\mathrm{adv},\kappa}; \tilde{\theta}^*)_i \ge \frac{1}{e^{l_\epsilon - l}} > 0$. However, $f(x_{\mathrm{adv},\kappa}; \tilde{\theta}^*)_i = 0$ by the definition of $\tilde{\theta}^*$ and $\mathcal{D}_{\mathrm{adv}}$. $\Rightarrow\Leftarrow$

$\square$

## C  EXPERIMENTAL DETAILS AND ADDITIONAL EXPERIMENTS

In this section, we provide a comprehensive account of the experiments corresponding to the tables and figures presented in the main paper, along with supplementary results and discussions that were excluded from the primary manuscript due to page constraints.

### C.1  DETAILED EXPERIMENTAL METHODOLOGY

For all experiments involving mean accuracy, mean corruption error (mCE), and adversarial robustness, we trained independent models on each benchmark three times and reported the average values. In the flatness experiments, we selected one of the trained models for each augmentation method to generate figures and calculate flatness metrics. Following the conventions from Hendrycks et al. (2021a), we used both the original data and augmented data in equal proportions across all experiments.

#### C.1.1  FLATNESS METRICS

**Descriptions on Metrics.** The maximum eigenvalue of Hessian matrix ($\lambda_{\mathrm{max}}$) represents locally steep or sharp direction in the loss landscape, implying that small changes in the parameters in this direction may lead to large changes in the loss. The trace of the Hessian (trace(H)) provides an overall measure of the curvature in all directions. $\mu_{\mathrm{PAC\text{-}Bayes}}$ represents simplified PAC-Bayesian bound introduced by Jiang et al. (2020). The simplified PAC-Bayesian bound, denoted as $\mu_{\mathrm{PAC\text{-}Bayes}}$, is computed as $1/\sigma$, where $\sigma$ is the largest value such that $|\mathcal{E}_{\mathcal{D}}(\theta + N(0, \sigma^2 I)) - \mathcal{E}_{\mathcal{D}}(\theta)| \leq \tau$. Therefore, flatter local minima correspond to larger $\sigma$ values and smaller $\mu_{\mathrm{PAC\text{-}Bayes}}$. The LPF-based flatness measure evaluates the flatness of a local minimum $\theta^*$ by computing the convolution of the loss function $\mathcal{L}$ with a Gaussian kernel $K = N(0, \sigma^2 I)$. Formally, it is defined as $(\mathcal{L} * K)(\theta^*) = \int \mathcal{L}(\theta^* - \tau)K(\tau)d\tau$. This measure effectively averages the loss over a neighborhood around $\theta^*$, with the Gaussian kernel weighting nearby points more heavily, thereby quantifying how sharp the loss landscape is in that region. $\epsilon$-sharpness ($\epsilon_{\mathrm{sharp}}$) measures the sensitivity of a loss function near a local minimum by quantifying the largest perturbation of model parameters that leads to an increase in loss of more than $\epsilon$.

**Method Implementations.** We have deployed power iteration (PI) method and Hutchinson's method to estimate $\lambda_{\mathrm{max}}$ and trace(H) with 20 iterations and 10 samples, respectively. We have adopted the default parameters for the $\mu_{\mathrm{PAC\text{-}Bayes}}$, LPF measure, and $\epsilon_{\mathrm{sharp}}$ i.e. $\tau = 0.05$, $\sigma = 0.05$, and $\epsilon = 0.1$ respectively.

**Loss Visualizations.** We visualized the loss surface using the same method as in Cha et al. (2021), but with a modification: we selected $\theta_2$ and $\theta_3$ as random orientation vectors of length 15. Specifically, given a trained model parameter $\theta_1$, we defined a two-dimensional orthogonal basis $u$ and $v$ as follows. We set $u = \theta_2 - \theta_1$, and computed $v$ by orthogonalizing $\theta_3 - \theta_1$ with respect to $u$, resulting in:

$$u = \theta_2 - \theta_1, v = \frac{(\theta_3 - \theta_1) - (\theta_3 - \theta_1)^\top (\theta_2 - \theta_1)}{\|\theta_2 - \theta_1\|^2 \cdot (\theta_2 - \theta_1)}.$$

Using this orthogonal basis, we created a two-dimensional grid and calculated the loss at each coordinate to visualize the loss surface.

#### C.1.2  ROBUSTNESS BENCHMARK DETAILS

**Domain Generalization Benchmarks.** We conducted comprehensive domain generalization experiments on the PACS, VLCS, and OfficeHome datasets to evaluate model robustness and generalizability using augmentations. These datasets are specifically designed to test a model's ability to generalize across different domains, as they consist of images that share the same labels but differ in their domain representations. For example, the PACS dataset contains images from four distinct domains—'art painting', 'cartoon', 'photo', and 'sketch'—all featuring the same classes such as 'dog', 'elephant', and others. In all experiments, models were trained until convergence to ensure fair and consistent comparisons across different settings.

Our baseline employed Empirical Risk Minimization (ERM) with the best hyperparameters selected from DomainBed's ERM training configuration (Gulrajani & Lopez-Paz, 2021). To specifically

assess the impact of augmentations on model performance, we excluded the heavy augmentation compositions used in the original DomainBed configuration.

For the baseline augmentations — including AugMix, PixMix, RandAug, StyleAug, and CutOut — we systematically tuned the number of training epochs, exploring options of 1,000, 2,500, and 5,000 epochs. The default setting was 5,000 epochs, aligning with the standard ERM training procedure. Specifically for StyleAug, we performed additional tuning of its hyperparameter $\alpha$ alongside the training epochs. Through manual experimentation, we found that setting $\alpha = 1.0$ yielded the best results in terms of model robustness and generalization performance.

**Common Corruption Benchmarks.** We assessed the robustness of models trained with augmentations against common corruptions using the CIFAR-10, CIFAR-100, and *tiny*ImageNet datasets. Specifically, we utilized their corrupted counterparts — CIFAR-10-C, CIFAR-100-C, and *tiny*ImageNet-C — which are standard benchmarks generated from the original datasets by applying 15 distinct corruption types at 5 different severity levels. These corruption types include brightness changes, contrast alterations, defocus blur, elastic transformations, fog addition, frost addition, Gaussian blur, glass distortion, impulse noise, JPEG compression, motion blur, pixelation, shot noise, snow addition, and zoom blur. Models were trained and validated on the respective clean datasets, with robustness evaluations performed at test time only to simulate real-world scenarios where models encounter unforeseen corruptions. To quantify model performance across all corruptions and severity levels, we calculated the mean Corruption Error (mCE), which averages the error rates over the different conditions.

For training, we adopted the configurations from Hendrycks et al. (2021b), with a modification to the maximum number of training epochs—extended from 100 to 400—to ensure thorough convergence. Specifically, we utilized stochastic gradient descent (SGD) with a momentum of 0.9 and a weight decay of 0.0005. A cosine learning rate decay schedule was applied over the 400 epochs to optimize learning efficiency.

Regarding the augmentation techniques, default configuration parameters were used for all baseline augmentations unless specified otherwise. For StyleAug, the hyperparameter $\alpha$ was set to 0.95 for the CIFAR experiments and adjusted to 1.0 for the TinyImageNet experiments to account for dataset-specific characteristics. In the CIFAR-10 experiment, CutOut was configured with a grid size of 4 to effectively augment the training data. All other augmentations — AugMix, PixMix, and RandAug — were applied using their default settings as described in their respective original works.

**Adversarial Robustness Benchmarks.** To further evaluate the robustness of models in adversarial settings, we conducted tests using untargeted Projected Gradient Descent (PGD) attacks based on the $L_\infty$ norm. For clarity, we utilize the absolute value $\alpha$ to represent gradient ascent coefficient, rather than a relative step size with respect to the perturbation magnitude $\eta$ (where $\eta$ corresponds to the commonly used epsilon $\epsilon$ in adversarial literature, representing the maximum allowed perturbation).

For the CIFAR-10 and CIFAR-100 experiments, we employed a PGD-7 attack with parameters $(\eta, \alpha) = (8/255, 2/255)$. In the *tiny*ImageNet experiment, we used a PGD-3 attack with parameters $(\eta, \alpha) = (3/255, 1/255)$. Essentially, more intense attacks have been applied to simpler datasets, while milder attacks have been used for more complex datasets, with the CIFAR experiments' attack configurations borrowed from the Yang et al. (2020). We then evaluate the adversarial robustness of models trained following the common corruption evaluation protocol detailed above.

## C.2 SUPPLEMENTARY EXPERIMENTAL RESULTS

This section contains supplementary results that were excluded from the primary manuscript due to page constraints.

### C.2.1 ADDITIONAL RESULTS ON ADVERSARIAL ROBUSTNESS

This section provides supplementary results, including adversarial accuracy (Table 5) for PGD $L_\infty$ attacks as detailed in C.1.2. Additionally, we present results for PGD $L_2$ untargeted attacks performed using a PGD-20 attack configuration with parameters $(\eta, \alpha) = (0.5, 1/800)$ (Table 6, 7). We also include the clean accuracies (Table 8) corresponding to each experimental result. Finally, Table 9 presents the preliminary results of our Theorem 5 concerning adversarial training using FGSM.

Table 5: Impact of augmentations with label smoothing on adversarial robustness. Error is computed against PGD $L_\infty$ untargeted attacks. 'Original' shows cross-entropy loss with original augmentation; '+ Label Smoothing (LS)' shows the loss with label smoothing applied. ($\downarrow$: *The lower the better).*

| Aug. | CIFAR-10 w/ $L_\infty$ $\downarrow$ | | CIFAR-100 w/ $L_\infty$ $\downarrow$ | | *tiny*ImageNet w/ $L_\infty$ $\downarrow$ | | Avg. Gain $\downarrow$ |
|---|---|---|---|---|---|---|---|
| | Original | + LS | Original | + LS | Original | + LS | |
| AugMix | 99.59 | 69.01 ($-30.58$) | 99.88 | 93.36 ($-6.52$) | 99.78 | 99.48 ($-0.30$) | **$-12.47$** |
| PixMix | 98.77 | 79.47 ($-19.30$) | 99.59 | 97.79 ($-1.80$) | 99.74 | 99.20 ($-0.54$) | **$-7.88$** |
| RandAug | 99.80 | 59.80 ($-40.00$) | 99.95 | 96.49 ($-3.46$) | 99.76 | 99.52 ($-0.24$) | **$-14.57$** |
| StyleAug | 99.89 | 68.04 ($-31.85$) | 99.96 | 96.99 ($-2.97$) | 100.00 | 99.96 ($-0.04$) | **$-11.62$** |
| CutOut | 99.96 | 62.80 ($-37.16$) | 99.97 | 97.23 ($-2.74$) | 99.66 | 99.47 ($-0.19$) | **$-13.36$** |

Table 6: Cross-entropy losses of different augmentations and their combination with label smoothing against adversarial perturbations from PGD $L_2$ untargeted attacks. 'Original' means the loss with original augmentation without label smoothing; '+ Label Smoothing (LS)' indicates the loss with label smoothing applied. ($\downarrow$: *The lower the better).*

| Aug. | CIFAR-10 w/ $L_2$ $\downarrow$ | | CIFAR-100 w/ $L_2$ $\downarrow$ | | *tiny*ImageNet w/ $L_2$ $\downarrow$ | | Avg. Gain $\downarrow$ |
|---|---|---|---|---|---|---|---|
| | Original | + LS | Original | + LS | Original | + LS | |
| AugMix | 0.024 | 0.006 ($-0.018$) | 0.051 | 0.025 ($-0.026$) | 0.012 | 0.012 ($-0.000$) | **$-0.015$** |
| PixMix | 0.018 | 0.007 ($-0.011$) | 0.041 | 0.029 ($-0.012$) | 0.011 | 0.011 ($-0.000$) | **$-0.008$** |
| RandAug | 0.030 | 0.006 ($-0.024$) | 0.061 | 0.028 ($-0.033$) | 0.013 | 0.013 ($-0.000$) | **$-0.019$** |
| StyleAug | 0.035 | 0.006 ($-0.029$) | 0.068 | 0.029 ($-0.039$) | 0.017 | 0.019 ($+0.002$) | **$-0.022$** |
| CutOut | 0.034 | 0.005 ($-0.029$) | 0.064 | 0.029 ($-0.035$) | 0.013 | 0.013 ($-0.000$) | **$-0.021$** |

Table 7: Impact of augmentations with label smoothing on adversarial robustness. Error is computed against PGD $L_2$ untargeted attacks. 'Original' shows cross-entropy loss with original augmentation; '+ Label Smoothing (LS)' shows the loss with label smoothing applied. ($\downarrow$: *The lower the better).*

| Aug. | CIFAR-10 w/ $L_2$ $\downarrow$ | | CIFAR-100 w/ $L_2$ $\downarrow$ | | *tiny*ImageNet w/ $L_2$ $\downarrow$ | | Avg. Gain $\downarrow$ |
|---|---|---|---|---|---|---|---|
| | Original | + LS | Original | + LS | Original | + LS | |
| AugMix | 63.89 | 29.09 ($-34.80$) | 92.76 | 75.83 ($-16.93$) | 60.54 | 59.01 ($-1.53$) | **$-17.75$** |
| PixMix | 59.11 | 36.01 ($-23.10$) | 89.75 | 83.62 ($-6.13$) | 57.17 | 56.26 ($-0.91$) | **$-10.05$** |
| RandAug | 72.53 | 27.25 ($-45.28$) | 95.11 | 80.67 ($-14.44$) | 61.89 | 59.55 ($-2.34$) | **$-20.02$** |
| StyleAug | 77.68 | 27.45 ($-50.23$) | 97.19 | 81.91 ($-15.28$) | 82.18 | 84.45 ($+2.27$) | **$-21.08$** |
| CutOut | 78.52 | 25.80 ($-52.72$) | 96.19 | 82.42 ($-13.77$) | 60.17 | 59.18 ($-0.99$) | **$-22.49$** |

Table 8: Clean error of different augmentations and their combination with label smoothing. 'Original' means the loss with original augmentation without label smoothing; '+ Label Smoothing (LS)' indicates the loss with label smoothing applied.

| Aug. | CIFAR-10 | | CIFAR-100 | | *tiny*ImageNet | |
|---|---|---|---|---|---|---|
| | Original | + LS | Original | + LS | Original | + LS |
| AugMix | 4.35 | 4.71 | 23.07 | 22.75 | 30.00 | 31.17 |
| PixMix | 3.72 | 4.89 | 21.39 | 23.38 | 29.49 | 30.89 |
| RandAug | 4.08 | 4.31 | 22.07 | 21.69 | 29.90 | 30.31 |
| StyleAug | 9.99 | 4.67 | 34.77 | 20.85 | 49.41 | 57.26 |
| CutOut | 3.77 | 4.69 | 21.50 | 21.15 | 29.85 | 30.24 |

### C.2.2 VISUALIZATIONS FOR DISTRIBUTIONS OF AUGMENTED SAMPLES

While augmentations have been developed for a variety of purposes, we have observed that those conforming to Assumption 1 generally exhibit superior generalization robustness compared to those that do not. Specifically, as shown in Tables 2 and 3, methods like AugMix, PixMix, and RandAug consistently enhance performance across diverse robustness benchmarks. These augmentations offer rich representations close to the original image, as illustrated in Figure 4. In contrast, StyleAug and

Table 9: Performance results of adversarial training (AT) using FGSM with $\epsilon = 2/255$. Our findings continue to align with Theorem 5.

| Aug. | CIFAR-10 ↓ | | CIFAR-100 ↓ | |
|---|---|---|---|---|
| | Original | + LS | Original | + LS |
| $L_2$ (loss) | 0.005 | 0.004 (−0.001) | 0.012 | 0.010 (−0.002) |
| $L_\infty$ (loss) | 0.026 | 0.010 (−0.016) | 0.046 | 0.033 (−0.013) |
| $L_2$ (acc) | 21.92 | 20.63 (−1.29) | 53.87 | 52.31 (−1.56) |
| $L_\infty$ (acc) | 64.61 | 44.90 (−19.71) | 92.11 | 87.34 (−4.77) |

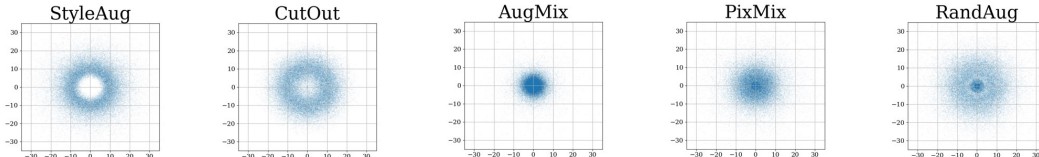

Figure 4: 2D representations of the augmentation distributions relative to the originals in the input space on CIFAR-100 dataset.

CutOut, which demonstrate marginal or low improvements on common corruption benchmarks, have sparse representations near the original image, resulting in minimal or no gains in robustness.

# D    FURTHER DISCUSSIONS

## D.1    WHAT ARE THE DESIRED ASPECTS OF AUGMENTATIONS?

Based on the theoretical foundations and the empirical investigations in our paper, we aim to figure out what are the desired aspects of augmentations. Revisiting Assumption 1, we found that the distribution of augmented samples, $\mathcal{P}_\mathcal{A}$, should cover the surrounding area of each original sample. By doing so, models aim to suppress the loss values within the area, leading to the flat and minimized loss surface around each input, which ultimately leads to the flatter surface in the parameter space.

However, in practice, augmentations do not have to densely encompass the surrounding region of each original sample. It is due to the fact that modern deep models with appropriate regularization methods show a strong interpolation performance on the input space. Therefore, we believe when augmentations sufficiently cover the wide range of surrounding region of each sample, deep models would show the flatter loss surface on the input space, thus on the parameter space.

However, not much existing augmentations are designed to fulfil our key lesson. As a further work, it is convincing to propose augmentation methods that are tailored to cover the surrounding region of each sample, e.g., additive Gaussian within a sphere, and combine it with the existing strong augmentation practices, e.g., reinforcement learning or generative model-based augmentations or Mixup techniques, to boost up the generalization performance of augmentations.

## D.2    HOW DO OUR CLAIMS FURTHER EXTEND TO LARGE & DEEP MODELS?

As one of the key limitations of our work, the proposed theory is applied to linear models, which is quite shallow than modern architectures. Specifically, the part involving the translation between input and parameter space, i.e., Theorem 1 and 2, are for linear models. The reason of this condition is we can further achieve the closed-form or tighter bounds of the solutions, i.e., $\mathcal{R}_\mathcal{X}^\gamma$, and $\mathcal{R}_\Theta^\gamma$, by using the linear architecture of given models.

However, in practice, we further investigate the proofs of our claims by using modern deep architectures, such as the WideResNet architecture. Based on the experiments which consistently verify our claim, we strongly believe that the main claims of this work can be extended to large and deep model architectures. For a more precise description, formalizing the regions seems to be intractable for complicated large and deep models. However, as a further insight, when a complex model is

assumed to be a continuous and smooth function (as widely accepted in deep machine learning theory), we speculate that our intuition, which links input/parameter regions, is also valid.

### D.3 ANY OTHER MODALITIES BEYOND VISUAL DATA?

Our main experiments handle visual data, more specifically, image classification tasks. Also, it might be unexplored that the key claims of this work can be further applied to other modalities, such as language, audio, or even sensory signals from devices. Our experiments seem to be limited to vision cases, but we believe that it is an important step forward to understand theoretical importance of augmentations in machine learning. Furthermore, we want to emphasize that our main theoretical claims do not explicitly rely on the characteristics of visual data, so it can be further extended to the other form of modalities. In addition, extending to other complicated task beyond classification can be an appropriate future research direction. In particular, sequential training such as natural language processing or decision based on reinforcement learning probably show distinct and unique theoretical understanding beyond classification.

