# OpenReview forum: "How Do Augmentations with Label Smoothing Enhance Model Robustness?"
_ICLR.cc/2025/Conference — Submitted to ICLR 2025_

### Official Review · Reviewer_GFi4 · 2024-10-29

**Soundness:** 3
**Presentation:** 3
**Contribution:** 2
**Rating:** 5
**Confidence:** 3

**Summary:**

This paper presents a theoretical understanding using linear models on how data augmentation relates to parameter space and therefore how data augmentation leads to flatter local mimima which are known to improve generalisation.

**Strengths:**

1. Clarity: This paper presents a nice theoretical framework with linear models on how augmentations relate to parameter spaces and thereby demonstrates that augmentation leads to a flatter local minima. They then validate that different types of augmentations do indeed lead to local minima on Cifar10 and other datasets.

2. Results: This work provides a formal framework to relate augmentations and parameter spaces that other work can build on. It formalises the intuition that augmentations / label smoothing indeed smooth the parameter space / predictions of the model.

**Weaknesses:**

1. The work is limited in being applied to only linear models and while the authors note that they believe it can be useful for explaining more complex models, I am hesitant as to the scope of applicability for two reasons:
a. Will conclusions transfer -- the authors look at this by considering datasets such as Cifar10 / etc. and showing that augmentation helps there. But this isn't the full story -- augmentations are not uniformly useful nor equally useful. Can such an understanding shed light on which ones would transfer more than others? Are there other conclusions that could be obtained and transferred? The fact that augmentations lead to smoother minima / generalisation is a known experimental fact [1] but I'm not sure that this formalisation helps further ones intuition
b. Do the insights give new ideas / a way of thinking that can improve larger models ? (Similar to point (a) but more generally.)

2. The experiments are not very convincing to me. The authors demonstrate known properties (e.g. augmentation improves on common corruptions) that the have demonstrated on linear models. What would be more convincing is if they could demonstrate that they can predict some property of augmentation and how that relates to the smoothness and thereby generalizability. Or that more smoothness leads to  more generalisation. Looking at Table 1 and 4 there doesn't seem to be much correlation.

[1] https://arxiv.org/pdf/1609.04836

**Questions:**

The paper is in general clear; I have not checked the math / proofs thoroughly.

If the authors could respond to my weaknesses above that would be helpful in case I have misunderstood their aims / the core of their experiments.

---

### Official Review · Reviewer_39fW · 2024-11-01

**Soundness:** 2
**Presentation:** 3
**Contribution:** 2
**Rating:** 3
**Confidence:** 4

**Summary:**

This paper presents a theoretical framework that explains how combining data augmentations with label smoothing enhances generalization and robustness. The authors establish a duality between the input space and model parameter space, showing that for each perturbation in the input space, corresponding perturbations exist in the parameter space that yield the same loss. According to their proposed theorem, data augmentation effectively flattens the loss landscape around the minima in parameter space, leading to improved generalization and robustness. The findings hold even when label smoothing is applied in conjunction, further boosting adversarial robustness.

**Strengths:**

* The motivation to offer a unified theoretical understanding of how data augmentation and label smoothing enhance generalization and robustness is strong.

* The paper is well-written, featuring clear organization and well-designed figures that support the presented concepts.

**Weaknesses:**

* The benefits of augmentation and label smoothing for improving generalization and robustness have been empirically studied and validated in numerous prior works. This paper’s primary contribution is theoretical; however, the proof of the proposed theorem is constrained to a bijective linear model, which is not fully representative of general deep learning scenarios where bijectivity and linearity are not guaranteed. In broader cases, there is no assurance that a perturbation in the input space will have a corresponding perturbation in the parameter space that maintains consistent loss (as stated in Theorem 1 and 2). When considering saddle points within the space, changes in the input space may increase the loss, while changes in the parameter space could decrease it. The proposed theorem will not hold in those cases.

* The experimental design for evaluating adversarial robustness lacks comprehensiveness; the evaluation metric should focus on accuracy under various strong attacks rather than solely comparing cross-entropy loss values. A lower cross-entropy loss does not necessarily indicate increased robustness. Conversely, a robust model might yield higher loss on specific samples while still preventing adversaries from finding perturbations within budget that effectively influence the model’s output.

**Questions:**

* It would be insightful for the authors to discuss or prove whether the theorems extend to general deep models with non-bijective activation functions (e.g. ReLU), or to explore any potential adjustments needed for deep models with common non-bijective activations.

* In section 3.3.3 evaluation results, did authors trained WRN-40-2 model on CIFAR-10-C and CIFAR-100-C or evaluated on them?

* The authors select several representative augmentation methods in their experiments, demonstrating that each technique contributes to a flatter parameter surface and improved generalization. However, the connection between these methods appears somewhat weak. It would be valuable to explore how adjusting augmentation parameters impacts the results. For instance, examining how changes in the severity parameter within RandAug affect the flatness of the loss surface could provide deeper insights.

* What is the adversarial and clean accuracy for Table 4 under the evaluation of AutoAttack?

* Do authors consider the impact of augmentation alone on adversarial robustness? And does label smoothing alone impact the flatness of loss surface and generalization ability?

---

### Official Review · Reviewer_jWuG · 2024-11-02

**Soundness:** 3
**Presentation:** 2
**Contribution:** 3
**Rating:** 6
**Confidence:** 2

**Summary:**

The paper provides a theoretical framework to illustrate why the data augmentation, label smoothing and their combination can generally improve the model’s out-of-distribution generalization performance and adversarial robustness of the model.

**Strengths:**

1 The paper provides solid mathematical proofs for each theorem presented.

2 The paper presents plausible theoretical studies to illustrate how data augmentation and label smoothing work effectively.

3 The comprehensive experiments have been conducted to support the findings.

**Weaknesses:**

1 The authors use numerous different mathematical notations, which makes it difficult to follow the meaning of each term. I personally suggest providing more figures for your theorem part

2 Readers without a strong mathematical background may find it challenging to understand the theorems provided.

**Questions:**

n/a

---

### Official Review · Reviewer_kueD · 2024-11-02

**Soundness:** 2
**Presentation:** 3
**Contribution:** 2
**Rating:** 3
**Confidence:** 3

**Summary:**

This paper presents a theoretical framework to analyze the relationship between data augmentations, label smoothing, model generalization, and adversarial robustness. The paper first proves that the augmentations can improve the model’s flatness. Further, the authors prove that combining the data augmentations and the label smoothing can enhance adversarial robustness. The paper also shows comprehensive empirical results to validate the correctness of the proposed theories.

**Strengths:**

1.	It seems that the authors provide a good explanation for why label smoothing can improve adversarial robustness.
2.	The authors provide empirical results to support their theoretical claims.

**Weaknesses:**

1.	The contribution seems limited. Admittedly, this paper aims to help explain the empirical phenomenon that label smoothing improves adversarial robustness. However, I do not think the only explanation is enough to be a paper accepted by a top-tier conference. It would be better to provide more insights about how to improve the label smoothing or the data augmentations to gain improved robustness from the theoretical view.
2.	The theoretical analyses make a strong assumption that the considered model is a linear model, which degrades the practicality of the theoretical results. In practice, adversarial training is always applied to a deep model instead of the linear model. It is unknown whether the theoretical results can still stand under the assumption of non-linear models.
3.	The empirical results seem to repeat the conclusions of the previous papers. The paper of data augmentations has already shown that their methods can improve generalization. Thus, there is no necessity to present such results again.

**Questions:**

Please refer to my comments in Weakness.

---

### Meta-Review · Area_Chair_ZfGJ · 2024-12-18

**Metareview:**

This work provides a unified theoretical framework linking data augmentations and label smoothing to improved model robustness by analyzing their effects on loss surface flatness, generalization bounds, and adversarial robustness, supported by extensive evaluations on corruption, adversarial, and domain generalization benchmarks. However, all reviewers expressed concerns that the technical contributions are limited and that the experiments are not comprehensive enough to support the claims made in the work. These concerns were not adequately addressed during the rebuttal, and therefore, we have decided not to accept this work in its current state.

**Additional Comments On Reviewer Discussion:**

N/A

---

### Decision · Program_Chairs · 2025-01-22

Reject